# Oracle-Efficient Combinatorial Semi-Bandits

**Jung-hun Kim**
CREST, ENSAE, IP Paris
FairPlay joint team, France
`junghun.kim@ensae.fr`

**Milan Vojnović**
London School of Economics
United Kingdom
`m.vojnovic@lse.ac.uk`

**Min-hwan Oh**
Seoul National University
South Korea
`minoh@snu.ac.kr`

## Abstract

We study the combinatorial semi-bandit problem where an agent selects a subset of base arms and receives individual feedback. While this generalizes the classical multi-armed bandit and has broad applicability, its scalability is limited by the high cost of combinatorial optimization, requiring oracle queries at *every* round. To tackle this, we propose oracle-efficient frameworks that significantly reduce oracle calls while maintaining tight regret guarantees. For the worst-case linear reward setting, our algorithms achieve $\widetilde{O}(\sqrt{T})$ regret using only $O(\log \log T)$ oracle queries. We also propose covariance-adaptive algorithms that leverage noise structure for improved regret, and extend our approach to general (non-linear) rewards. Overall, our methods reduce oracle usage from linear to (doubly) logarithmic in time, with strong theoretical guarantees.

## 1 Introduction

The combinatorial semi-bandit problem extends the classical multi-armed bandit (MAB) model to settings where an agent selects a subset of base arms (a combinatorial action) and receives individual feedback for each. This general framework captures many real-world scenarios, such as product recommendation, where a set of items is recommended to a user [16]; ad slot allocation, where multiple ads are displayed on a webpage [12]; and network routing, where a path comprising several links is selected in a communication network [27].

Due to its broad applicability, the combinatorial semi-bandit problem has been extensively studied in the literature [5, 7, 18, 9, 22, 32]. However, a central challenge lies in the computational complexity of solving the combinatorial optimization problem, which is often NP-hard. As a result, most existing algorithms assume access to an oracle that returns a solution to the combinatorial problem. These algorithms rely on querying the oracle at every round, leading to excessive oracle usage and substantial computational overhead in practice.

In this work, following the computational complexity notions introduced in Balkanski and Singer [1], Fahrbach et al. [11], we distinguish between two measures of oracle efficiency: *adaptivity complexity* and *query complexity*, which are defined later. Our goal is to improve oracle efficiency by substantially reducing the overall oracle adaptivity and query complexities in decision-making over a time horizon $T$, while maintaining tight gap-free regret guarantees that do not depend on the suboptimality gaps. Our main contributions are summarized below and compared with prior work on gap-free combinatorial semi-bandits in Table 1.

- **Oracle-efficient algorithms for worst-case linear rewards:** We propose two frameworks that significantly reduce oracle query usage while maintaining tight regret guarantees. Using an adaptive oracle query framework, `AROQ-CMAB` achieves near-optimal regret of $\widetilde{O}(\sqrt{mdT})$, with both adaptivity and query complexity bounded by $O(d \log \log(Tm/d))$, where $d$ denotes the number of base arms and $m$ is the maximum number of activated base arms per action. To further improve computational practicality by reducing adaptivity complexity,

39th Conference on Neural Information Processing Systems (NeurIPS 2025).

Table 1: Gap-free regret bounds for combinatorial semi-bandit algorithms.

| Combinatorial Reward Model | Algorithm | Regret | Adaptivity Complexity | Query Complexity |
|---|---|---|---|---|
| Linear (Worst-case) | CUCB [5] | $\widetilde{O}(m\sqrt{dT})$ | $\Theta(T)$ | $\Theta(T)$ |
| | CUCB [18] | $\widetilde{O}(\sqrt{mdT})$ | $\Theta(T)$ | $\Theta(T)$ |
| | AROQ-CMAB (**our work**) | $\widetilde{O}(\sqrt{mdT})$ | $O(d\log\log(\frac{Tm}{d}))$ | $O(d\log\log(\frac{Tm}{d}))$ |
| | SROQ-CMAB (**our work**) | $\widetilde{O}(m\sqrt{dT})$ | $\Theta(\log\log T)$ | $O(d\log\log T)$ |
| Linear (Covariance-dependent) | OLS-UCB-C [32] | $\widetilde{O}\left(\sqrt{\sum_{i\in[d]}\max_{a\in\mathcal{A}\text{ s.t. }i\in a}\sigma_i^2(a)T}\right)$ | $\Theta(T)$ | $\Theta(T)$ |
| | AROQ-C-CMAB (**our work**) | $\widetilde{O}\left(\sqrt{\sum_{i\in[d]}\max_{a\in\mathcal{A}\text{ s.t. }i\in a}\sigma_i^2(a)T}\right)$ | $O(d^2\log(Tm))$ | $O(d^2\log(Tm))$ |
| | SROQ-C-CMAB (**our work**) | $\widetilde{O}\left(\sqrt{d\max_{a\in\mathcal{A}}\sum_{i\in a}\sigma_i^2(a)T}\right)$ | $\Theta(\log\log T)$ | $O(d^2\log\log T)$ |
| General | SDCB [4] | $\widetilde{O}(L\sqrt{mdT})$ | $\Theta(T)$ | $\Theta(T)$ |
| | AROQ-GR-CMAB (**our work**) | $\widetilde{O}(L\sqrt{mdT})$ | $O(d\log\log(\frac{Tm}{d}))$ | $O(d\log\log(\frac{Tm}{d}))$ |
| | SROQ-GR-CMAB (**our work**) | $\widetilde{O}(Lm\sqrt{dT})$ | $\Theta(\log\log T)$ | $O(d\log\log T)$ |

we propose a scheduled oracle query framework that executes multiple independent oracle queries in parallel. Under this framework, SROQ-CMAB achieves regret $\widetilde{O}(m\sqrt{dT})$ with adaptivity complexity of $\Theta(\log\log T)$ and query complexity of $O(d\log\log T)$.

- **Covariance-adaptive oracle-efficient algorithms for linear rewards:** Utilizing our proposed frameworks, we design oracle-efficient algorithms that leverage the estimated covariance structure of the reward noise. AROQ-C-CMAB achieves near-optimal regret $\widetilde{O}\big(\sqrt{\sum_{i\in[d]}\max_{a\in\mathcal{A}\text{ s.t. }i\in a}\sigma_i^2(a)T}\big)$, where $\sigma_i^2(a)$ denotes the variance contribution of base arm $i$ under action $a$, with adaptivity and query complexities of $O(d^2\log(Tm))$. SROQ-C-CMAB achieves regret $\widetilde{O}\big(\sqrt{d\max_{a\in\mathcal{A}}\sum_{i\in a}\sigma_i^2(a)T}\big)$ with adaptivity complexity of $\Theta(\log\log T)$ and query complexity of $O(d^2\log\log T)$.

- **Oracle-efficient algorithm for general reward models:** We extend our frameworks to general (non-linear) reward functions. AROQ-GR-CMAB achieves regret $\widetilde{O}(L\sqrt{mdT})$ with adaptivity and query complexities of $O(d\log\log(Tm/d))$, where $L$ denotes the maximum possible value of the reward. SROQ-GR-CMAB achieves regret $\widetilde{O}(Lm\sqrt{dT})$ with adaptivity complexity of $O(\log\log T)$ and query complexity of $\Theta(d\log\log T)$.

**Related Work.** The combinatorial semi-bandit problems have been extensively studied, starting from the foundational work of Chen et al. [5]. Kveton et al. [18] established tight regret bounds that are near-optimal. Further improvements were made by Combes et al. [7], who derived better bounds under the assumption that the feedback from selected arms is independent.

More recently, a unified framework that accounts for both dependent and independent feedback through covariance analysis was introduced by Degenne and Perchet [9], assuming knowledge of the covariance matrix. This line of research has been further advanced by Perrault et al. [22] and Zhou et al. [32], who developed covariance-adaptive algorithms based on confidence ellipsoids. In addition to linear reward structures, generalized linear reward functions have also been studied in the combinatorial semi-bandit setting by Chen et al. [4].

Despite these advances, all of the aforementioned works require solving a combinatorial optimization problem frequently at every round using an offline oracle, which is generally NP-hard [8]. To alleviate the computational burden, Cuvelier et al. [8] proposed an approximation-based approach that achieves polynomial-time complexity. However, their method introduces a trade-off between regret and computational cost, as achieving near-optimal regret necessitates increasingly accurate

approximations—leading to potentially unbounded computational time.[1] Similarly, Chen et al. [5] considered approximation oracles but focused on minimizing approximate regret, rather than the original regret, and their method is limited to cases where such approximation oracles are available. Neu and Bartók [20] studied efficient algorithms in the adversarial semi-bandit setting, but their method still requires solving the optimization problem at every round and does not attain optimal regret in the stochastic setting. Similarly, Zhou et al. [32] proposed an adaptive covariance-based algorithm using ellipsoidal confidence regions, yet it also incurs oracle calls at every round. Lastly, Tzeng et al. [26] studied matroid semi-bandits with sublinear per-round computational complexity by exploiting matroid structure, a direction that is complementary to our focus on reducing the number of oracle calls for combinatorial semi-bandits with arbitrary action sets.

As a related line of research, oracle-efficient algorithms have been studied for submodular function optimization problems [1, 2, 11]. However, these approaches do not involve latent models that can be learned from stochastic sequential feedback, as in bandit learning. As a result, they differ fundamentally in formulation and are not applicable to the bandit setting. Oracle-efficient bandit algorithms have also been proposed for bandit linear optimization [17], achieving $O(\text{poly}(d, \log T))$ oracle complexity. However, these approaches assume linear rewards with full-arm decisions and do not handle combinatorial action spaces or semi-bandit feedback, which are central to our setting. To the best of our knowledge, rare oracle queries in combinatorial semi-bandit problems have only been empirically explored by Combes et al. [7], who proposed a heuristic using $O(\log T)$ oracle calls but without theoretical regret guarantees. Furthermore, several variants of combinatorial bandits heavily rely on frequent oracle queries, including Thompson Sampling methods [28], maximum-reward feedback settings [29], and pure exploration problems [3].

## 2 Problem Formulation

There are $d$ base arms, and let $\mathcal{A} \subseteq \{0, 1\}^d$ denote the set of available actions, where each action $a \in \mathcal{A}$ is a binary vector indicating the activated base arms. We allow $\mathcal{A}$ to be an arbitrary subset of $\{0, 1\}^d$. Then, we define $m = \max_{a \in \mathcal{A}} \|a\|_0$ as the maximum number of activated base arms across all actions. At each time $t \in [T]$, the environment samples a vector of rewards $y_t \in [0, 1]^d$ from a fixed distribution $\mathcal{D}$ that is unknown to the agent, and the agent chooses an action $a_t \in \mathcal{A}$. For any vector $x \in \mathbb{R}^d$, we use $x_i$ to denote its $i$-th entry. Then, the agent receives a reward $r(a_t, y_t)$ where $r : \mathcal{A} \times [0, 1]^d \to \mathbb{R}$, and observes the values of $y_{t,i}$ for each $i \in [d]$ such that $a_{t,i} = 1$ (semi-bandit feedback). The mean of the latent distribution $\mathcal{D}$ is denoted by $\mu = (\mu_1, \ldots, \mu_d)$. We first focus on the standard linear reward setting studied in prior work [18, 7, 9, 22, 32], where the reward is given by $r(a, y_t) := \langle a, y_t \rangle$. We will discuss generalizations beyond the linear case later.

**Regret.** Let $a^*$ be an optimal action, $a^* \in \text{argmax}_{a \in \mathcal{A}} \bar{r}(a)$, where $\bar{r}(a) = \mathbb{E}_{y \sim \mathcal{D}}[r(a, y)]$ represents the expected reward function (e.g., $\bar{r}(a) = \langle a, \mu \rangle$ in the linear case). The goal is to minimize the cumulative regret over horizon $T$, defined as $\mathcal{R}(T) = \mathbb{E}[\sum_{t=1}^{T} (\bar{r}(a^*) - \bar{r}(a_t))]$.

**Combinatorial Optimization.** For finding an optimal action, it is required to solve the combinatorial optimization problem $\text{argmax}_{a \in \mathcal{A}} \bar{r}(a)$, whose computational cost, in general with arbitrary $\mathcal{A} \subseteq \{0, 1\}^d$, is proportional to the size of $\mathcal{A}$ which is $O(d^m)$. To address this computational complexity, the previous work on combinatorial semi-bandits [7, 5, 18] assumed access to an oracle, which returns a solution for the combinatorial optimization. However, these methods require querying the oracle at every round. In this work, we aim to substantially reduce the number of oracle queries while achieving tight regret. Formally, as in the previous work, we assume access to an oracle that returns $a^\dagger \in \text{argmax}_{a \in \mathcal{A}} f(a)$ for given $f : \mathcal{A} \to \mathbb{R}$. Furthermore, our oracle-efficient approach can be incorporated with an approximation oracle, which will be discussed later.

**Oracle Efficiency.** Following the computational complexity notions introduced in [1, 11], we evaluate the oracle efficiency of our algorithms using two key measures, which are described as follows and illustrated in Figure 1: **Query complexity** refers to the total number of *individual* oracle queries made over the entire time horizon. This reflects the standard computational workload of the

---

[1]To reach optimal regret, the approximation level $\delta_t$ must satisfy $\lim_{t \to \infty} \delta_t = 0$, and each round incurs cost $O(1/\delta_t) \to \infty$

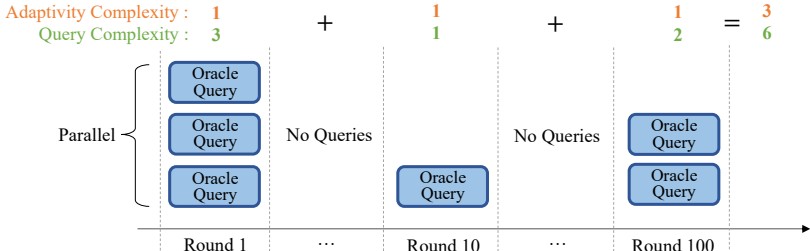

Figure 1: Oracle adaptivity and query complexities.

algorithm. **Adaptivity complexity** captures the number of *sequential* rounds of oracle queries, where each round may consist of a set of queries that can be executed in parallel without depending on each other.

While query complexity is important in general, adaptivity complexity often dominates the actual runtime cost in parallel or distributed environments, where parallelizing queries within a round is easy, but synchronizing between rounds incurs overhead. Our goal is to reduce both measures of oracle complexity while maintaining tight regret guarantees.

**Additional Notation.** With a slight abuse of notation, we write $i \in a$ for $a \in \mathcal{A}$ if the $i$-th coordinate of $a$ satisfies $a_i = 1$. For any $b, c \in \mathbb{R}$, we write $b \lesssim c$ to indicate that $b$ is upper bounded by $c$ up to a constant factor. We use $b_+$ for $\max\{b, 0\}$.

## 3 Oracle-Efficient Algorithms and Regret Analysis

We present two frameworks for combinatorial semi-bandit problems with rare oracle queries: one based on an *adaptive* oracle calls, and the other based on *scheduled* oracle queries, both inspired by batch learning [21, 25, 13, 6, 10, 31, 15, 23, 30, 14].

### 3.1 Adaptive Rare Oracle Queries

We propose an algorithm (Algorithm 1) that leverages adaptive oracle queries. The algorithm employs a UCB-based strategy with adaptive epoch-based updates, enabling efficient exploration despite infrequent oracle access. We use the UCB indices defined as: for some constant $C > 0$,

$$r_t^{UCB}(a) = \sum_{i \in a} \left( \hat{\mu}_{t,i} + \sqrt{\frac{C \log t}{n_{t,i}}} \right), \tag{1}$$

where $\hat{\mu}_{t,i} = (1/n_{t,i}) \sum_{s=1}^{t-1} y_{s,i} \mathbb{1}(i \in a_s)$ and $n_{t,i} = \sum_{s=1}^{t-1} \mathbb{1}(i \in a_s)$. The UCB index for each action is updated when there exists a base arm $i$ whose selection count exceeds a specified threshold. More precisely, the indices are updated when the number of rounds in which arm $i$ has been selected in the current epoch, denoted by $|\mathcal{T}_i(\tau_i)|$, satisfies $|\mathcal{T}_i(\tau_i)| \geq 1 + \sqrt{Tm \cdot |\mathcal{T}_i(\tau_i - 1)|/d}$, where $\tau_i$ is the epoch index for arm $i$, and $\mathcal{T}_i(\tau)$ is the set of rounds in epoch $\tau$ where $i$ was selected.

The intuition for the threshold condition of oracle queries is as follows: for $i \in [d]$ with $\tau_i$, the instance regret is bounded by $\sqrt{1/|\mathcal{T}_i(\tau_i - 1)|}$ so that with the bound for $|\mathcal{T}_i(\tau_i)|$ from the update condition, the overall regret for $\tau_i$ epochs is bounded by $|\mathcal{T}_i(\tau_i)|\sqrt{1/|\mathcal{T}_i(\tau_i - 1)|} \lesssim \sqrt{Tm|\mathcal{T}_i(\tau_i - 1)|/d}\sqrt{1/|\mathcal{T}_i(\tau_i - 1)|} = \sqrt{Tm/d}$. By considering all $\tau_i$ for $i \in [d]$, we can obtain the near-optimal regret bound with oracle efficiency.

**Theorem 1** *With oracle adaptivity and query complexities of $O(d \log \log(Tm/d))$, respectively, Algorithm 1 achieves a regret bound of*

$$\mathcal{R}(T) = O\left( \sqrt{mdT \log T} \log \log(Tm/d) \right).$$

*Proof.* The full version of the proof is provided in Appendix A.2. ∎

---

**Algorithm 1** Adaptive Rare Oracle Queries for Combinatorial MAB (`AROQ-CMAB`)

---

**Initialize:** $\tau_i = 1$ for all $i \in [d]$
**for** $t = 1, 2..., T$ **do**
    **for** $i \in [d]$ *s.t.* $|\mathcal{T}_i(\tau_i)| \geq 1 + \sqrt{Tm \cdot |\mathcal{T}_i(\tau_i - 1)|/d}$ **do**
        $\tau_i \leftarrow \tau_i + 1, \mathcal{T}_i(\tau_i) \leftarrow \emptyset$
        $Update \leftarrow True$
    **if** $Update = True$ **then**
        $a_t \leftarrow \arg\max_{a \in \mathcal{A}} r_t^{UCB}(a)$ with (1)                            *// Oracle Query*
        $Update \leftarrow False$
    **else**
        $a_t \leftarrow a_{t-1}$
    Play $a_t$ and observe feedback $y_{t,i}$ for $i \in a_t$
    $\mathcal{T}_i(\tau_i) \leftarrow \mathcal{T}_i(\tau_i) \cup \{t\}$ for all $i \in a_t$

---

**Comparison to Previous Work.** The worst-case regret lower bound for this problem is known to be $\Omega(\sqrt{mdT})$ [5], and a near-optimal regret was achieved by the algorithm of Chen et al. [5], Kveton et al. [18]. However, their approach requires $\Theta(T)$ oracle adaptivity and query complexity, respectively. In contrast, our proposed algorithm, `AROQ-CMAB` (Algorithm 1), achieves a near-optimal regret bound while significantly reducing both oracle adaptivity and query complexities to $O(d \log \log(Tm/d))$.

**$\alpha$-Approximation Oracle.** Prior work on combinatorial bandits [5] mitigates the cost of exact oracles by using $\alpha$-approximation oracles, which return an action $a^\dagger$ satisfying $f(a^\dagger) \geq \alpha \max_{a \in \mathcal{A}} f(a)$ for some approximation factor $\alpha \in [0, 1]$, when such oracles are available. This leads to analyzing the regret relative to the best $\alpha$-approximate reward, rather than the true optimal reward. In contrast, our method targets the original (non-approximate) regret while reducing the frequency of oracle queries. Nevertheless, our framework can naturally incorporate $\alpha$-approximation oracles, achieving the same regret guarantee in terms of $\alpha$-approximate regret with only infrequent approximate oracle calls. See Appendix A.3 for details.

### 3.2 Scheduled Rare Oracle Queries

To further reduce adaptivity complexity and enable more efficient distributed computation, we propose an algorithm that performs scheduled batched oracle queries at predetermined epochs (Algorithm 2), allowing these oracle queries to be synchronized and executed in parallel. At each epoch $\tau$, we define UCB and LCB indices as: for some constant $C > 0$

$$r_\tau^{UCB}(a) = \sum_{i \in a} \left( \hat{\mu}_{\tau,i} + \sqrt{\frac{C \log T}{n_{\tau,i}}} \right) \text{ and } r_\tau^{LCB}(a) = \sum_{i \in a} \left( \hat{\mu}_{\tau,i} - \sqrt{\frac{C \log T}{n_{\tau,i}}} \right), \quad (2)$$

where $\hat{\mu}_{\tau,i} = (1/n_{\tau,i}) \sum_{t=1}^{t_\tau - 1} y_{t,i} \mathbb{1}(i \in a_t)$, $n_{\tau,i} = \sum_{t=1}^{t_\tau - 1} \mathbb{1}(i \in a_t)$, and $t_\tau$ denotes the start time of the $\tau$-th epoch.

To schedule oracle queries, we adopt an elimination-based bandit strategy [25]. However, in combinatorial bandits, the exponentially large number of suboptimal actions poses a significant challenge for efficient exploration. To address this, we construct a representative action $a_\tau^{(i)}$ for each base arm $i \in [d]$ in epoch $\tau$. Using elimination conditions applied to these representative actions, we can efficiently eliminate a large number of suboptimal actions by focusing on suboptimal base arms—those that are not part of the optimal action $a^*$. In each epoch, these representative actions are selected for exploration, after which the estimators and representative actions are updated. This process requires oracle queries but only at the batch level, keeping the query frequency low.

Let $\mathcal{T} = \{t_1, \ldots, t_M\}$ denote the set of time steps at which oracle queries are made, where $M > 0$, $t_1 = 1$, $t_M = T$, and for $1 < \tau < M$, the sequence is defined recursively as $t_\tau = \eta \sqrt{t_{\tau-1}}$. The scaling factor is set to $\eta = T^{1/(2-2^{1-M})}$. We choose $M = \Theta(\log \log T)$ to ensure doubly logarithmic adaptivity and query complexity. Parallel execution of oracle queries is discussed in Appendix A.1.

---

**Algorithm 2** Scheduled Rare Oracle Queries for Combinatorial MAB (`SROQ-CMAB`)

---
**Input:** $\mathcal{T}$
1 **for** $\tau = 1, 2, \ldots, M$ **do**
2    Update $\hat{\mu}_\tau := (\hat{\mu}_{\tau,1}, \ldots, \hat{\mu}_{\tau,d})$
3    $a_\tau^{(i)} := \mathrm{argmax}_{a \in \mathcal{A}_{\tau-1}: i \in a}\, r_\tau^{UCB}(a)$ for all $i \in \mathcal{N}_{\tau-1}$ with (2)      *// Oracle Queries*
4    $\mathcal{N}_\tau \leftarrow \{i \in \mathcal{N}_{\tau-1} \mid r_\tau^{UCB}(a_\tau^{(i)}) \geq \max_{a \in \mathcal{A}_{\tau-1}} r_\tau^{LCB}(a)\}$ with (2)      *// Oracle Query*
5    $\mathcal{A}_\tau \leftarrow \{a \in \mathcal{A}_{\tau-1} \mid a_i = 0 \text{ for all } i \in [d]/\mathcal{N}_\tau\}$
6    $\mathcal{T}_\tau \leftarrow [t_\tau, t_{\tau+1} - 1]$
7    **for** $t \in \mathcal{T}_\tau$ **do**
8      $i \leftarrow \big((t-1) \bmod |\mathcal{N}_\tau| + 1\big)$-th element of $\mathcal{N}_\tau$.
9      Play $a_t = a_\tau^{(i)}$ and observe feedback $y_{t,i}$ for $i \in a_t$

---

**Theorem 2** *With oracle adaptivity complexity of $\Theta(\log \log T)$ and oracle query complexity of $O(d \log \log T)$, Algorithm 2 achieves a regret bound of*

$$\mathcal{R}(T) = O\left(m\sqrt{dT \log T \log \log T}\right).$$

*Proof.* The full version of the proof is provided in Appendix A.4. ∎

We observe that Algorithm 2 improves the adaptivity complexity from $O(d \log \log(Tm/d))$ in Algorithm 1 to $\Theta(\log \log T)$, at the cost of an additional $\sqrt{m}$ factor in the regret.

**Remark 1** *In practice, Algorithm 2 can be more computationally efficient than Algorithm 1. First, due to its reduced adaptivity complexity, Algorithm 2 enables more efficient parallel execution of oracle queries. Second, the elimination process progressively discards suboptimal base arms, reducing the oracle query complexity per round from $O(d^m)$ to $O(|\mathcal{N}_\tau|^m)$, where $|\mathcal{N}_\tau| \leq d$ denotes the number of remaining base arms at epoch $\tau$. These computational advantages are further supported by our experimental results presented later.*

Our proposed frameworks for combinatorial semi-bandits with rare oracle queries can be extended to variants of the combinatorial semi-bandit, including covariance-dependent CMAB and general-reward CMAB. In the following, we examine each of these settings in turn.

## 4    Extension to Covariance-dependent CMAB

In this section, instead of targeting worst-case regret, we consider covariance-dependent regret, inspired by [9, 22, 32]. The covariance-dependent analysis can cover independent or dependent (worst-case) semi-bandit rewards of arms in an action. Here, the covariance matrix for the reward distribution $\mathcal{D}$ is denoted by $\Sigma \in \mathbb{R}^{d \times d}$, which is assumed to be unknown to the agent.

For simplicity, we assume that for any $1 \leq i \leq j \leq d$, there exists an action $a \in \mathcal{A}$ such that $a_i = a_j = 1$. When this assumption does not hold, our algorithms and analysis extend naturally by restricting attention to *observable pairs*, that is, pairs $(i, j)$ for which there exists an action $a \in \mathcal{A}$ with $a_i = a_j = 1$. All results continue to hold under this extension, with the same regret guarantees.

In the following, we propose covariance-adaptive algorithms based on our two frameworks—adaptive and scheduled rare oracle queries—to handle this setting.

### 4.1    Adaptive Rare Oracle Queries for Covariance-adaptive Approach

We first propose an algorithm (Algorithm 3) based on the adaptive rare oracle query framework. Recall that $n_{t,i} = \sum_{s=1}^{t-1} \mathbb{1}(i \in a_s)$ and $n_t := (n_{t,1}, \ldots, n_{t,d})$. Let $D_x$ and $D_X$ denote diagonal matrices, where $D_x$ has the entries of vector $x$ on its diagonal, and $D_X$ has the diagonal entries of matrix $X$.

We define the estimated means as $\hat{\mu}_t = D_{n_t}^{-1} \sum_{s=1}^{t-1} D_{a_s} y_s$. We also define covariance estimator $\hat{\Sigma}_t = \hat{S}_t - \hat{\mu}_t \hat{\mu}_t^\top$ where $\hat{S}_{t,(i,j)} = \frac{1}{n_{t,(i,j)}} \sum_{s=1}^{t-1} a_{s,i} a_{s,j} y_{s,i} y_{s,j}$, and confidence bound $\overline{\Sigma}_{t,(i,j)} =$

---

**Algorithm 3** Adaptive Rare Oracle Queries for Covariance-adaptive CMAB (`AROQ-C-CMAB`)

---
**Initialize:** $\tau_{i,j} = 0$ for all $i,j \in [d] \times [d]$
**for** $t = 1, 2..., T$ **do**
    **if** $t \le \lceil d(d+1)\log^3(T)/2 \rceil$ **then**
        Let $(i,j)$ be the $\left((t-1) \bmod \frac{d(d+1)}{2} + 1\right)$-th pair in a fixed enumeration of all pairs $(i,j)$
        with $1 \le i \le j \le d$
        $a_t \leftarrow$ any $a \in \mathcal{A}$ s.t. $i \in a, j \in a$
        **if** $t = \lceil d(d+1)\log^3(T)/2 \rceil$ **then**
            $\tau_{i,j} \leftarrow \tau_{i,j} + 1$ for all $i,j \in [d] \times [d]$
    **else**
        **for** $i,j \in [d] \times [d]$ s.t. $|\mathcal{T}_{i,j}(\tau_{i,j})| \ge 1 + 2|\mathcal{T}_{i,j}(\tau_{i,j} - 1)|$ **do**
            $\tau_{i,j} \leftarrow \tau_{i,j} + 1, \mathcal{T}_{i,j}(\tau_{i,j}) \leftarrow \emptyset$
            $Update \leftarrow True$
        **if** $Update = True$ **then**
            $a_t \leftarrow \arg\max_{a \in \mathcal{A}} r_t^{UCB}(a)$ with (3)                    // Oracle Query
            $Update \leftarrow False$
        **else**
            $a_t \leftarrow a_{t-1}$
    Play $a_t$ and observe feedback $y_{i,t}$ for $i \in a_t$
    $\mathcal{T}_{i,j}(\tau_{i,j}) \leftarrow \mathcal{T}_{i,j}(\tau_{i,j}) \cup \{t\}$ for all $i \in a_t$ and $j \in a_t,$

---

$\hat{\Sigma}_{t,(i,j)} + \frac{1}{4}\left(\frac{5h_t}{\sqrt{n_{t,(i,j)}}} + \frac{h_t^2}{n_{t,(i,j)}} + \frac{1}{n_{t,(i,j)}^2}\right)$, where $n_{t,(i,j)} = \sum_{s=1}^{t-1} \mathbb{1}(i \in a_s)\mathbb{1}(j \in a_s)$ with $n_{t,(i,i)} = n_{t,i}$, and $h_t = O(\sqrt{\log t + \log d})$. Define the gram matrix $\overline{G}_t = \sum_{s=1}^{t-1} D_{a_s}\overline{\Sigma}_t D_{a_s} + D_{\overline{\Sigma}_t} D_{n_t} + I$. Then, we utilize the UCB index, defined as

$$r_t^{UCB}(a) = \langle a, \hat{\mu}_t \rangle + f_t \|D_{n_t}^{-1} a\|_{\overline{G}_t}, \tag{3}$$

where $f_t = O(\log t + d \log\log t)$.

To initialize, the algorithm uniformly explores actions as a warm-up phase for the stability of the covariance estimator. Then in the main stage, for the adaptive update condition, we adopt a stricter criterion than that of (worst-case) Algorithm 1, resulting in more frequent updates—though oracle queries remain rare—while handling the covariance-dependent bound to achieve a tighter guarantee. For the covariance dependent regret bound, we define $\sigma_i^2(a) = \sum_{j \in a}(\Sigma_{i,j})_+$. The algorithm achieves a near-optimal regret bound asymptotically as follows.

**Theorem 3** *With oracle adaptivity and query complexities of $O(d^2 \log(Tm))$, respectively, Algorithm 3 achieves an asymptotic regret bound of*

$$\mathcal{R}(T) = \widetilde{O}\left(\sqrt{T\sum_{i \in [d]} \max_{a \in \mathcal{A} \text{ s.t. } i \in a} \sigma_i^2(a)}\right).$$

*Proof.* The full version of the proof is provided in Appendix A.5. ∎

In the worst case of dependent base arm rewards, our regret bound becomes $\widetilde{O}(\sqrt{mdT})$, which is the same as that of Algorithm 1. However, for the independent reward case across all base arms (i.e., $\Sigma = I$), the regret bound becomes $\widetilde{O}(\sqrt{dT})$, which is tighter by a factor of $\sqrt{m}$.

**Comparison to Previous Work.** As discussed in [32], the proposed algorithms in [9, 22] achieved gap-dependent asymptotic regret bound, which is not tight for the gap-free bound with respect to $T$ because of the additional $1/\Delta_{\min}^2$ factor. The regret lower bound of this problem is $\Omega(\sqrt{T\sum_{i \in [d]} \max_{a \in \mathcal{A} \text{ s.t. } i \in a} \sigma_i^2(a)})$ [32] and [32] propose an algorithm achieving near-optimal gap-free asymptotic regret bound of $\widetilde{O}(\sqrt{T\sum_{i \in [d]} \max_{a \in \mathcal{A} \text{ s.t. } i \in a} \sigma_i^2(a)})$ with $\Theta(T)$ oracle adaptivity

**Algorithm 4** Scheduled Rare Oracle Queries for Covariance-adaptive CMAB (SROQ-C-CMAB)

**Input:** $\mathcal{T}$

10 **for** $t \in [1, \lceil d(d+1)/2 \rceil]$ **do**

11 $\quad$ Let $(i,j)$ be the $\left((t-1) \bmod \frac{d(d+1)}{2} + 1\right)$-th pair in a fixed enumeration of all pairs $(i,j)$ with $1 \le i \le j \le d$

12 $\quad$ $a_t \leftarrow$ any $a \in \mathcal{A}$ s.t. $i \in a, j \in a$

13 **for** $\tau = 1, 2, \ldots, M$ **do**

14 $\quad$ Update $\hat{\mu}_{\tau,i}, \hat{S}_{\tau,(i,j)}$ for all $(i,j) \in [d] \times [d]$

15 $\quad$ $a_\tau^{(i)} := \mathrm{argmax}_{a \in \mathcal{A}_{\tau-1}:i \in a} r_\tau^{UCB}(a)$ for all $i \in \mathcal{N}_{\tau-1}$ with (4) $\hfill$ *// Oracle Queries*

16 $\quad$ $a_\tau^{(i,j)} := \mathrm{argmax}_{a \in \mathcal{A}_{\tau-1}:i,j \in a} r_\tau^{UCB}(a)$ for all $(i,j) \in \mathcal{N}_{\tau-1}^{(2)} : i \ne j$ $\hfill$ *// Oracle Queries*

17 $\quad$ $\mathcal{N}_\tau \leftarrow \{i \in \mathcal{N}_{\tau-1} \mid r_\tau^{UCB}(a_\tau^{(i)}) \ge \max_{a \in \mathcal{A}_{\tau-1}} r_\tau^{LCB}(a)\}$ with (4) $\hfill$ *// Oracle Query*

18 $\quad$ $\mathcal{A}_\tau' \leftarrow \{a \in \mathcal{A}_{\tau-1} \mid a_i = 0 \text{ for all } i \in [d]/\mathcal{N}_\tau\}$

19 $\quad$ $\mathcal{N}_\tau^{(2)} \leftarrow \{(i,j) \in \mathcal{N}_\tau \times \mathcal{N}_\tau \mid r_\tau^{UCB}(a_\tau^{(i,j)}) \ge \max_{a \in \mathcal{A}_\tau'} r_\tau^{LCB}(a), i \ne j\}$ $\hfill$ *// Oracle Query*

20 $\quad$ $\mathcal{A}_\tau \leftarrow \{a \in \mathcal{A}_\tau' \mid a_i = 0 \text{ or } a_j = 0 \text{ for all } (i,j) \in [d] \times [d]/\mathcal{N}_\tau^{(2)}, i \ne j\}$

21 $\quad$ $\mathcal{T}_\tau^{(1)} \leftarrow [t_\tau, t_\tau + T_\tau - (d^2 m^2 T_\tau \log T)^{2/3} - 1], \mathcal{T}_\tau^{(2)} \leftarrow [t_\tau + T_\tau - (d^2 m^2 T_\tau \log T)^{2/3}, t_{\tau+1} - 1]$

22 $\quad$ **for** $t \in \mathcal{T}_\tau^{(1)}$ **do**

23 $\quad\quad$ $i \leftarrow (t \bmod |\mathcal{N}_\tau|)$-th element in $\mathcal{N}_\tau$

24 $\quad\quad$ Play $a_t = a_\tau^{(i)}$

25 $\quad\quad$ Receive reward $\langle a_t, y_t \rangle$ and observe feedback $y_{t,i}$ for $i \in [d]$ s.t. $a_{t,i} = 1$

26 $\quad$ **for** $t \in \mathcal{T}_\tau^{(2)}$ **do**

27 $\quad\quad$ $(i,j) \leftarrow (t \bmod |\mathcal{N}_\tau^{(2)}|)$-th element in $\mathcal{N}_\tau^{(2)}$

28 $\quad\quad$ Play $a_t = a_\tau^{(i,j)}$

29 $\quad\quad$ Receive reward $\langle a_t, y_t \rangle$ and observe feedback $y_{t,i}$ for $i \in [d]$ s.t. $a_{t,i} = 1$

and query complexity, respectively. Our algorithm achieves the near-optimal asymptotic regret bound with reduced oracle adaptivity and query complexity of $O(d^2 \log(mT))$, respectively.

### 4.2 Scheduled Rare Oracle Queries for Covariance-adaptive Approach

Here, we propose a covariance-adaptive algorithm (Algorithm 4) by utilizing the framework of scheduled rare oracle queries. Recall that for each epoch $\tau$, $\hat{\mu}_{\tau,i} = (1/n_{\tau,i}) \sum_{t=1}^{t_\tau - 1} y_{t,i} \mathbb{1}(i \in a_t)$ where $n_{\tau,i} = \sum_{t=1}^{t_\tau - 1} \mathbb{1}(i \in a_t)$ and $t_\tau$ is the start time of epoch $\tau$ in the algorithm. For the covariance estimator, we define $\hat{\Sigma}_\tau = \hat{S}_\tau - \hat{\mu}_\tau \hat{\mu}_\tau^\top$ where $\hat{S}_{\tau,(i,j)} = (1/n_{\tau,i,j}) \sum_{t=1}^{t_\tau - 1} a_{t,i} a_{t,j} y_{t,i} y_{t,j}$, and confidence bound $\overline{\Sigma}_{\tau,(i,j)} = \hat{\Sigma}_{\tau,(i,j)} + \frac{1}{4}\left(\frac{5h_T}{\sqrt{n_{\tau,(i,j)}}} + \frac{h_T^2}{n_{\tau,(i,j)}} + \frac{1}{n_{\tau,(i,j)}^2}\right)$, where $n_{\tau,(i,j)} = \sum_{t=1}^{t_\tau - 1} \mathbb{1}(i \in a_t)\mathbb{1}(j \in a_t), n_{\tau,(i,i)} = n_{\tau,i}$, and $h_T = O(\sqrt{\log T + \log d})$. Define the gram matrix $\overline{G}_\tau = \sum_{s=1}^{t_\tau - 1} D_{a_s} \overline{\Sigma}_\tau D_{a_s} + D_{\overline{\Sigma}_\tau} D_{n_\tau} + I$. Then, for the confidence bounds, we utilize

$$r_\tau^{UCB}(a) = \langle a, \hat{\mu}(\tau) \rangle + f_T \|D_{n_\tau}^{-1} a\|_{\overline{G}_\tau} \text{ and } r_\tau^{LCB}(a) = \langle a, \hat{\mu}(\tau) \rangle - f_T \|D_{n_\tau}^{-1} a\|_{\overline{G}_\tau}, \quad (4)$$

where $f_T = O(\log T + d \log \log T)$. For scheduled oracle queries, we employ the same time grid $\mathcal{T} = \{t_1, \ldots, t_M\}$, as in Algorithm 2, shifted by $\lceil d(d+1)/2 \rceil$ time steps to account for the warm-up phase, that is, $t_k \leftarrow t_k + \lceil d(d+1)/2 \rceil$ for all $k \in [M]$. Parallel execution of oracle is discussed in Appendix A.1.

**Theorem 4** *With oracle adaptivity complexity of $\Theta(\log \log T)$ and oracle query complexity of $O(d^2 \log \log T)$, Algorithm 4 achieves an asymptotic regret bound of*

$$\mathcal{R}(T) = \widetilde{O}\left(\sqrt{d \max_{a \in \mathcal{A}} \sum_{i \in a} \sigma_i^2(a) T}\right).$$

*Proof.* The full version of the proof is provided in Appendix A.6. ■

In the worst case, the scheduled query-based Algorithm 4 achieves a regret bound of $\widetilde{O}(m\sqrt{dT})$, which matches that of the scheduled query-based Algorithm 2 but is larger than the bounds achieved by the adaptive oracle query framework of Algorithm 1 and Algorithm 3 by a factor of $\sqrt{m}$. This gap arises from the inefficiency of using a fixed-time framework compared to an adaptive-time framework. However, the oracle complexities of Algorithm 4 are significantly lower than the $O(d^2 \log(mT))$ complexities of Algorithm 3, achieving $\Theta(\log \log T)$ adaptivity and $O(d^2 \log \log T)$ query complexity.

## 5 Extension to General-Reward CMAB

In this section, beyond a linear reward function we explored in the previous sections, we consider general reward functions $r(a, y)$ defined on $\mathcal{A} \times [0, 1]^d \to [0, L]$ for $L > 0$. We adopt the same setting with assumptions as in [4]. Specifically, at each time $t$, the arms produce stochastic outcomes $(y_{t,i})_{i=1}^d \in [0, 1]^d$ drawn i.i.d. over time from a distribution $\mathcal{D}$ with finite support[2] and the expected reward is denote by $\bar{r}(a) = \mathbb{E}_{y \sim \mathcal{D}}[r(a, y)]$. In this setting, we consider the following assumption.

**Assumption 1 (Monotone reward function)** *For any $y, y' \in [0, 1]^d$ satisfying $y_i \leq y'_i$ for all $i \in [d]$ and any $a \in \mathcal{A}$, we have $r(a, y) \leq r(a, y')$.*

The assumption for the monotone reward function is commonly observed in various combinatorial problems such as $K$-MAX [24], $K$-SUM [5], and Expected Utility Maximization [19].

By adopting our adaptive and scheduled frameworks for rare oracle queries, we propose oracle-efficient algorithms for general reward CMAB; Algorithms 6 and 7, respectively. The details of the algorithms are provided in Appendix A.7. In what follows, we provide theorems for oracle complexities and regret of each algorithm. The proofs are provided in Appendices A.8, A.9.

**Theorem 5** *With oracle adaptivity and query complexities of $O(d \log \log(Tm/d))$, respectively, Algorithm 6 achieves a regret bound of $\mathcal{R}(T) = \widetilde{O}(L\sqrt{dmT})$.*

**Theorem 6** *With oracle adaptivity complexity of $\Theta(\log \log T)$ and oracle query complexity of $O(d \log \log T)$, Algorithm 7 achieves a regret bound of $\mathcal{R}(T) = \widetilde{O}(Lm\sqrt{dT})$.*

**Comparison to Previous Work.** Chen et al. [4] proposed an algorithm achieving a regret bound of $\widetilde{O}(L\sqrt{mdT})$ with oracle adaptivity and oracle query complexities of $\Theta(T)$. However, our algorithms achieve $\widetilde{O}(L\sqrt{mdT})$ and $\widetilde{O}(Lm\sqrt{dT})$, respectively, requiring significantly reduced oracle complexities of of order $\log \log T$.

## 6 Experiments

We compare our algorithms to benchmarks in terms of oracle efficiency and regret using synthetic datasets[3]. We begin with the linear reward setting, where the mean vector is sampled from Unif$[0, 1]$ with $d = 20$ and $m = 3$, and stochastic rewards are uniformly generated around these means at each round. As shown in Figure 2 (a,b), our algorithms (AROQ-CMAB, SROQ-CMAB) achieve significantly lower oracle adaptivity and query complexities than CUCB [5], consistent with Theorems 1 and 2. Importantly, as shown in Figure 2 (d), our algorithms achieve faster runtime than the benchmark. In particular, SROQ-CMAB outperforms AROQ-CMAB in runtime, benefiting from a lower total adaptivity complexity up to $T$ (Figure 2 (e)), which enables more efficient parallel oracle execution (Remark 1). Figure 2(c) demonstrates that AROQ-CMAB incurs slightly higher regret and SROQ-CMAB incurs somewhat larger regret than CUCB, which is consistent with our theoretical predictions: the regret bounds involve an additional logarithmic factor for AROQ-CMAB and a $\sqrt{m}$

---

[2]The finite-support assumption simplifies the algorithms and analysis but is not essential. As noted in Chen et al. [4], the results can extend to Lipschitz-continuous reward functions by discretization techniques that preserve the same regret bound (see Appendix A.10).

[3]Source Code: https://github.com/junghunkim7786/OracleEfficientCombinatorialBandits

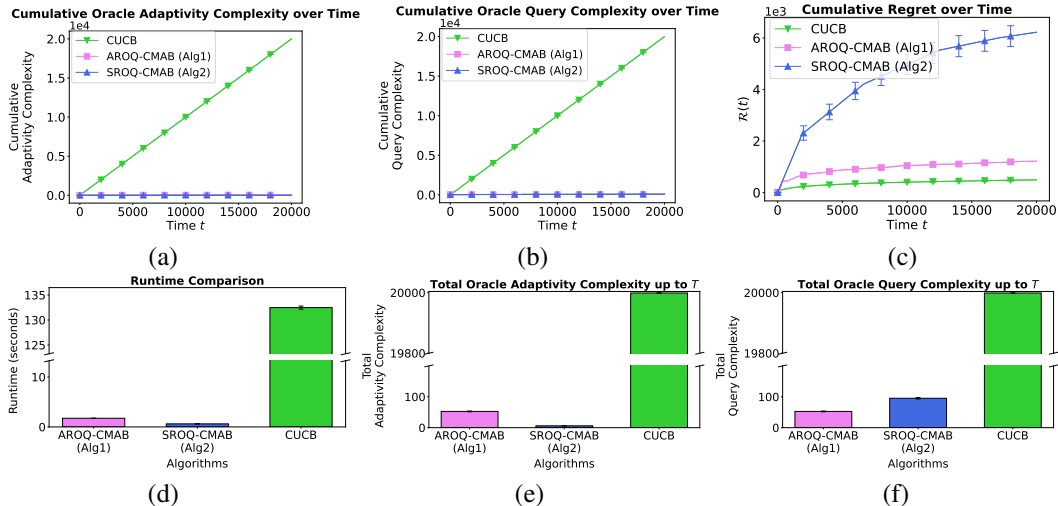

Figure 2: Experimental results for linear rewards with $d = 20$ and $m = 3$.

factor for `SROQ-CMAB`. Additional results for the covariance-adaptive variants and general reward functions are provided in Appendix A.11.

# 7    Conclusion

In this work, we proposed oracle-efficient algorithms for semi-combinatorial bandits. We introduced two algorithmic frameworks for handling rare oracle queries—adaptive and scheduled—and demonstrated that our algorithms significantly improve oracle efficiency while maintaining tight regret guarantees for worst-case linear rewards, covariance-dependent linear rewards, and general (non-linear) reward functions.

**Societal Impact.**    The research is primarily theoretical and does not engage with human subjects, sensitive data, or domains with identifiable risks of negative societal impact.

# Acknowledgements

J. Kim acknowledges the support of ANR through the PEPR IA FOUNDRY project (ANR-23-PEIA-0003) and the Doom project (ANR-23-CE23-0002), as well as the ERC through the Ocean project (ERC-2022-SYG-OCEAN-101071601). M. Oh was supported by the National Research Foundation of Korea (NRF) grant funded by the Korea government (MSIT) (No. RS-2022-NR071853 and RS-2023-00222663), by the Global-LAMP Program of the NRF grant funded by the Ministry of Education (No. RS-2023-00301976), and by AI-Bio Research Grant through Seoul National University.

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

# A    Appendix

## A.1    Details for Parallel Execution of Oracle Queries in the Scheduled Framework

This section provides further details on the parallel execution of oracle queries within the scheduled framework. Specifically, we describe how our proposed algorithms (Algorithms 2, 4, and 7) support *parallel execution of oracle queries* to reduce adaptivity complexity.

**Worst-Case Linear Rewards.**    In Algorithm 2, at the beginning of each epoch, the oracle queries in Line 3 can be executed in parallel. Moreover, although Line 4 depends on the result of Line 3, the oracle queries in Lines 3 and 4 can still be executed in parallel, as they involve independent computations. Specifically, evaluating $\max_{a \in \mathcal{A}} r_\tau^{\mathrm{LCB}}(a)$ in Line 4 requires only a single independent oracle call that returns the maximizer $a^\dagger = \mathrm{argmax}_{a \in \mathcal{A}} r_\tau^{\mathrm{LCB}}(a)$, allowing us to directly retrieve the maximum value $r_\tau^{\mathrm{LCB}}(a^\dagger)$.

**Covariance-Dependent Linear Rewards.**    In Algorithm 4, the oracle queries in Lines 15, 16, and 17 can be executed in parallel, following the same principle as in the worst-case linear rewards setting. In contrast, the query in Line 19 must be performed *sequentially afterward* since the oracle relies on the updated set $\mathcal{A}'_\tau$ resulting from the preceding computation.

**General Rewards.**    In Algorithm 7, following the same principle as in the worst-case linear rewards setting, the oracle queries in Lines 32 and 33 can be executed in parallel.

## A.2    Proof of Theorem 1

Let $\tau_i(t)$ be the value of $\tau_i$ at time $t$ in the algorithm. Then $\tau_i(T)$ represents the number of updates, each update requiring oracle queries, up to $T$ from arm $i$. Then, for the bound of the total oracle queries up to $T$, by adopting the proof techniques in [10], we have the following lemma.

**Lemma 1 (Oracle Queries Bound)**  *We have*

$$\mathbb{E}\left[\sum_{i \in [d]} \tau_i(T)\right] = O(d \log \log(Tm/d)).$$

*Proof.*    We first provide the following lemma.

**Lemma 2**  *For $M \geq 0$ and a sequence $x_0, x_1, \ldots$ such that $x_i \geq 1 + \sqrt{M x_{i-1}}$ for all $i \geq 1$, we have that $x_\tau \geq M^{1-2^{-\tau+1}}$ for all $\tau \geq 1$.*

*Proof.*    For $\tau = 1$, we have
$$x_1 \geq 1 + \sqrt{M x_0} \geq 1 = M^0,$$
which satisfies the desired inequality. We now proceed by induction. Suppose that for some $\tau \geq 1$, the inequality $x_\tau \geq M^{1-2^{-\tau+1}}$ holds. Then, using the recurrence, we have

$$\begin{aligned}
x_{\tau+1} &\geq 1 + \sqrt{M x_\tau} \\
&\geq 1 + \sqrt{M \cdot M^{1-2^{-\tau+1}}} \\
&= 1 + M^{(2-2^{-\tau+1})/2} \\
&= 1 + M^{1-2^{-\tau}} \\
&\geq M^{1-2^{-\tau}}.
\end{aligned}$$

Thus, by induction, we conclude that

$$x_\tau \geq M^{1-2^{-\tau+1}} \quad \text{for all } \tau \geq 1.$$

$\blacksquare$

Let $\tau_0 = \log \log(Tm/d)$. From Lemma 2, if $\tau \geq \tau_0 + 1$ and $\tau$ is not the last stage, for any $i \in [d]$, we have

$$n_{\tau,i} \geq (Tm/d)^{1-2^{-\tau+1}} \geq (Tm/d)^{1-2^{-\log\log(Tm/d)}} = Tm/2d.$$

Therefore, from the fact that $\sum_{t=1}^{T} \|a_t\|_0 \leq mT$, there are at most $2d + d$ (including the last stages for all $i \in [d]$) pairs of $(i, \tau)$ for $i \in [d]$ and $\tau \in [\tau_i(T)]$ satisfying $\tau \geq \tau_0 + 1$. This implies that for $i \in [d]$ s.t. $\tau_i(T) \geq \tau_0 + 1$, we have $\sum_{i \in [d]} \mathbb{1}(\tau_i(T) \geq \tau_0 + 1)\tau_i(T) \leq d\tau_0 + 3d$. Therefore, we have

$$\mathbb{E}\left[\sum_{i \in [d]} \tau_i(T)\right] \leq d\tau_0 + \mathbb{E}\left[\sum_{i \in [d]} \mathbb{1}(\tau_i(T) \geq \tau_0 + 1)\tau_i(T)\right] \lesssim d\tau_0 = d \log\log(Tm/d),$$

which concludes the proof. ∎

For the proof of regret bound, we utilize the Azuma-Hoeffding inequality provided in the following lemma.

**Lemma 3 (Azuma-Hoeffding Inequality)** *For a martingale difference sequence $X_1, \ldots, X_n$ with support of size $1$ for all $X_i$, for $\gamma > 0$ we have*

$$\mathbb{P}\left(\sum_{i=1}^{n} X_i \geq \gamma n\right) \leq 2\exp(-2\gamma^2 n).$$

From Lemma 3, we can show that the event $\mathcal{E}_t = \{|\mu_i - \hat{\mu}_{s,i}| \leq \sqrt{\frac{1.5 \log s}{n_{s,i}}} \ \forall 1 \leq s \leq t \ \forall i \in [d]\}$ holds with probability of at least $1 - O(\frac{d}{t^2})$. For ease of presentation, we define auxiliary variables $n'_{t,i}$ and $\hat{\mu}'_{t,i}$ for each $i \in [d]$ and time step $t \in [T]$ for the analysis on the rare updated indexes as follows: If the selected action at time $t$ is newly updated in the algorithm (i.e., Update = True), then

$$n'_{t,i} = n_{t,i}, \qquad \hat{\mu}'_{t,i} = \hat{\mu}_{t,i}.$$

Otherwise, if the previously selected action is maintained (i.e., Update = False), then

$$n'_{t,i} = n'_{t-1,i}, \qquad \hat{\mu}'_{t,i} = \hat{\mu}'_{t-1,i}.$$

Using these adjusted statistics, we define the UCB-based optimistic reward estimate for any action $a$ as

$$\widetilde{r}_t^{\text{UCB}}(a) = \sum_{i \in a}\left(\hat{\mu}'_{t,i} + \sqrt{\frac{C \log t}{n'_{t,i}}}\right).$$

Now we provide a bound for the regret as follows:

$$\mathcal{R}(T) = \mathbb{E}\left[\sum_{t \in [T]} (\bar{r}(a^*) - \bar{r}(a_t))\mathbb{1}(\mathcal{E}_t)\right] + \mathbb{E}\left[\sum_{t \in [T]} (\bar{r}(a^*) - \bar{r}(a_t))\mathbb{1}(\mathcal{E}_t^c)\right]$$

$$\leq \mathbb{E}\left[\sum_{t \in [T]} (\bar{r}(a^*) - \bar{r}(a_t))\mathbb{1}(\mathcal{E}_t)\right] + O(dm)$$

$$\lesssim \mathbb{E}\left[\sum_{t \in [T]} (\widetilde{r}_t^{UCB}(a^*) - \bar{r}(a_t))\mathbb{1}(\mathcal{E}_t)\right]$$

$$\leq \mathbb{E}\left[\sum_{t \in [T]} (\widetilde{r}_t^{UCB}(a_t) - \bar{r}(a_t))\mathbb{1}(\mathcal{E}_t)\right]$$

$$\lesssim \mathbb{E}\left[\sum_{t \in [T]} \sum_{i \in a_t} \sqrt{\frac{\log T}{n'_{t,i}}}\right],$$

where the second and last inequalities are obtained from $\mathcal{E}_t$. For bounding the last term, we have

$$
\begin{aligned}
\mathbb{E}\left[\sum_{t\in[T]}\sum_{i\in a_t}\sqrt{\frac{\log T}{n'_{t,i}}}\right] &= \mathbb{E}\left[\sum_{i\in[d]}\sum_{\tau\in[\tau_i(T)]}\sum_{t\in\mathcal{T}_i(\tau)}\sqrt{\frac{\log T}{n'_{t,i}}}\right] \\
&\leq \mathbb{E}\left[\sum_{i\in[d]}\sum_{\tau\in[\tau_i(T)]}\sum_{t\in\mathcal{T}_i(\tau)}\sqrt{\frac{\log T}{|\mathcal{T}_i(\tau-1)|}}\right] \\
&= \mathbb{E}\left[\sum_{i\in[d]}\sum_{\tau\in[\tau_i(T)]}|\mathcal{T}_i(\tau)|\sqrt{\frac{\log T}{|\mathcal{T}_i(\tau-1)|}}\right] \\
&\lesssim \mathbb{E}\left[\sum_{i\in[d]}\sum_{\tau\in[\tau_i(T)]}\sqrt{\frac{Tm\cdot|\mathcal{T}_i(\tau-1)|}{d}}\sqrt{\frac{\log T}{|\mathcal{T}_i(\tau-1)|}}\right] \\
&= \mathbb{E}\left[\sum_{i\in[d]}\tau_i(T)\right]\sqrt{\frac{Tm\log T}{d}} \\
&\lesssim \log\log(Tm/d)\sqrt{dmT\log T},
\end{aligned}
$$

where the first inequality is obtained from $n'_{t,i} \geq |\mathcal{T}_i(\tau-1)|$ for $t\in\mathcal{T}_i(\tau)$, the second equality is obtained from the condition of updates in the algorithm, and the last inequality is obtained from Lemma 1.

**Oracle Complexity Bounds.** Based on Lemma 1, we can show that the oracle query complexity is bounded by $O(d\log\log(Tm/d))$. Since the epochs for each base arm are updated separately, the adaptivity complexity is also bounded by $O(d\log\log(Tm/d))$.

### A.3 $\alpha$-Approximation Oracle

In this section, we provide a detailed explanation for $\alpha$-approximation oracle. We focus on the adaptive rare oracle query framework, noting that similar results can be derived for the scheduled framework, which we omit to avoid redundancy. Instead of obtaining the exact solution, the $\alpha$-approximation oracle, denoted by $\mathbb{O}_t^\alpha$, outputs action $a_t^\alpha \in \mathcal{A}$ satisfying

$$
r_t^{UCB}(a_t^\alpha) \geq \alpha\max_{a\in\mathcal{A}}r_t^{UCB}(a)
$$

for $\alpha > 0$. We investigate the $\alpha$-regret, which is defined as

$$
\mathcal{R}^\alpha(T) = \mathbb{E}\left[\sum_{t\in[T]}\alpha\bar{r}(a^*) - \bar{r}(a_t)\right].
$$

---

**Algorithm 5** $\alpha$-approximated Adaptive Rare Oracle Queries for Combinatorial MAB ($\alpha$-AROQ-CMAB)

---
**Initialize:** $\tau_i = 1$ for all $i \in [N]$
**for** $t = 1, 2..., T$ **do**
  $a_t \leftarrow a_{t-1}$
  **for** $i \in [d]$ *s.t.* $|\mathcal{T}_i(\tau_i)| \geq 1 + \sqrt{Tm\cdot|\mathcal{T}_i(\tau_i-1)|/d}$ **do**
    $\tau_i \leftarrow \tau_i + 1, \mathcal{T}_i(\tau_i) \leftarrow \emptyset$
    $Update \leftarrow True$
  **if** $Update = True$ **then**
    $a_t \leftarrow \mathbb{O}_t^\alpha$
    $Update \leftarrow False$
  Play $a_t$ and observe feedback $y_{t,i}$ for $i \in a_t$
  $\mathcal{T}_i(\tau_i) \leftarrow \mathcal{T}_i(\tau_i) \cup \{t\}$ for all $i \in a_t$

---

**Theorem 7** *With oracle adaptivity and query complexities of $O(d \log \log(Tm/d))$, respectively, Algorithm 5 achieves a $\alpha$-regret bound of*

$$\mathcal{R}^\alpha(T) = O\left(\sqrt{mdT \log T} \log \log(Tm/d)\right).$$

*Proof.* Here we provide the part that is different from the proof of Theorem 1. We provide a bound for the regret as follows:

$$
\begin{aligned}
\mathcal{R}^\alpha(T) &= \mathbb{E}\left[\sum_{t \in [T]} (\alpha \bar{r}(a^*) - \bar{r}(a_t)) \mathbb{1}(\mathcal{E}_t)\right] + \mathbb{E}\left[\sum_{t \in [T]} (\alpha \bar{r}(a^*) - \bar{r}(a_t)) \mathbb{1}(\mathcal{E}_t^c)\right] \\
&\leq \mathbb{E}\left[\sum_{t \in [T]} (\alpha \bar{r}(a^*) - \bar{r}(a_t)) \mathbb{1}(\mathcal{E}_t)\right] + O(md) \\
&\lesssim \mathbb{E}\left[\sum_{t \in [T]} (\alpha \tilde{r}_t^{UCB}(a^*) - \bar{r}(a_t)) \mathbb{1}(\mathcal{E}_t)\right] \\
&\leq \mathbb{E}\left[\sum_{t \in [T]} (\tilde{r}_t^{UCB}(a_t) - \bar{r}(a_t)) \mathbb{1}(\mathcal{E}_t)\right] \\
&\lesssim \mathbb{E}\left[\sum_{t \in [T]} \sum_{i \in a_t} \sqrt{\frac{\log T}{n'_{t,i}}}\right],
\end{aligned}
$$

where the second last inequality is obtained from $\mathbb{O}_t^\alpha$. The other parts of the proof are the same as those of Theorem 1. $\blacksquare$

### A.4 Proof of Theorem 2

From Lemma 3, we can show that the event $\mathcal{E} = \{|\mu_i - \hat{\mu}_i(\tau)| \leq \sqrt{\frac{1.5 \log T}{n_{\tau,i}}} \; \forall \tau \in [M], \forall i \in [d]\}$ holds with probability of at least $1 - \frac{d}{T^2}$. Then by adopting the proof technique in [6], under the event $\mathcal{E}$, we show that $A_\tau$, activated arm set for $\tau$-th epoch, always contains optimal arm $a^*$ in the following lemma.

**Lemma 4** *Under $\mathcal{E}$, we can show that for all $\tau \in [M]$, $a^* \in \mathcal{A}_\tau$.*

*Proof.* This can be shown by induction. Suppose $a^* \in \mathcal{A}_\tau$ for fixed $\tau \in [M]$. Under $\mathcal{E}$, we have $\bar{r}(a) \leq r_{\tau+1}^{UCB}(a)$ and $\bar{r}(a) \geq r_{\tau+1}^{LCB}(a)$ for any $a \in \mathcal{A}_\tau$. Then for any fixed $i \in a^*$, for any $a \in \mathcal{A}_\tau$, we can show that

$$r_{\tau+1}^{UCB}(a_{\tau+1}^{(i)}) \geq r_{\tau+1}^{UCB}(a^*) \geq \bar{r}(a^*) \geq r^{LCB}(a),$$

where the first inequality is obtained from the definition of $a_{\tau+1}^{(i)}$ with $a^* \in \mathcal{A}_\tau$. This implies that

$$r_{\tau+1}^{UCB}(a_{\tau+1}^{(i)}) \geq \max_{a \in \mathcal{A}_\tau} r_{\tau+1}^{LCB}(a),$$

which implies that $i \in a^*$ is not eliminated from the elimination condition at the $\tau + 1$-th epoch. This holds for all $i \in a^*$ so that $a^* \in \mathcal{A}_{\tau+1}$. With $a^* \in \mathcal{A}_0 = \mathcal{A}$, we can conclude the induction. $\blacksquare$

Under $\mathcal{E}$, we have

$$R(a^*) - \bar{r}(a_\tau^{(i)}) \leq r_\tau^{LCB}(a^*) + 2 \sum_{j \in a^*} \sqrt{\frac{1.5 \log t_\tau}{n_{\tau,j}}} - r_\tau^{UCB}(a_\tau^{(i)}) + 2 \sum_{j \in a_\tau^{(i)}} \sqrt{\frac{1.5 \log t_\tau}{n_{\tau,j}}}$$

$$\lesssim \sum_{j \in a^*} \sqrt{\frac{\log t_\tau}{n_{\tau,j}}} + \sum_{j \in a_\tau^{(i)}} \sqrt{\frac{\log t_\tau}{n_{\tau,j}}}$$

$$\lesssim \sum_{j \in a^*} \sqrt{\frac{d \log t_\tau}{|\mathcal{T}_{\tau-1}|}} + \sum_{j \in a_\tau^{(i)}} \sqrt{\frac{d \log t_\tau}{|\mathcal{T}_{\tau-1}|}}$$

$$\lesssim m \sqrt{\frac{d \log t_\tau}{|\mathcal{T}_{\tau-1}|}},$$

where the first inequality is obtained from $\mathcal{E}$, the second inequality comes from the fact that $a^* \in \mathcal{A}_{\tau-1}$ from Lemma 4, and $\max_{a \in \mathcal{A}_{\tau-1}} r_\tau^{LCB}(a) \leq r_\tau^{UCB}(a_\tau^{(i)})$ from the algorithm, and the third inequality is obtained from $n_{\tau,i} \geq \sum_{t \in \mathcal{T}_{\tau-1}} \mathbb{1}(i \in a_t) \gtrsim |\mathcal{T}_{\tau-1}|/d$ from the exploration in the algorithm at the $\tau-1$-th epoch.

Finally, we can show that

$$\mathcal{R}(T) \leq \mathbb{E}\left[\sum_{\tau=1}^{M} \sum_{t \in \mathcal{T}_\tau} \bar{r}(a^*) - \bar{r}(a_t) \mid \mathcal{E}\right] \mathbb{P}(\mathcal{E}) + \mathbb{E}\left[\sum_{\tau=1}^{M} \sum_{t \in \mathcal{T}_\tau} \bar{r}(a^*) - \bar{r}(a_t) \mid \mathcal{E}^c\right] \mathbb{P}(\mathcal{E}^c)$$

$$\lesssim \mathbb{E}\left[\sum_{\tau=1}^{M} \sum_{t \in \mathcal{T}_\tau} \bar{r}(a^*) - \bar{r}(a_t) \mid \mathcal{E}\right] + \frac{md}{T^2}$$

$$\lesssim \sum_{\tau=1}^{M} |\mathcal{T}_\tau| m \sqrt{\frac{d \log T}{|\mathcal{T}_{\tau-1}|}}$$

$$\leq \sum_{\tau=1}^{M} \eta m \sqrt{d |\mathcal{T}_{\tau-1}| \log T}$$

$$= m \eta \sqrt{dMT \log T}$$

$$\lesssim m \sqrt{dT \log(T) \log \log(T)},$$

which concludes the proof for the regret bound.

**Oracle Complexity Bounds.** Based on the oracle calls in Lines 3 and 4 of Algorithm 2, we observe that each epoch involves at most $d$ independent oracle queries (Line 3) and one sequential oracle query (Line 4). Since the total number of epochs is $M = \Theta(\log \log T)$, the overall adaptivity complexity is bounded by $\Theta(\log \log T)$, and the query complexity is bounded by $O(d \log \log T)$.

### A.5 Proof of Theorem 3

Let $\tau_{i,j}(t)$ be the value of $\tau_{i,j}$ at time $t$ in the algorithm. Then $\tau_{i,j}(T)$ represents the number of updates, each update requiring Oracle queries, up to $T$ from a pair of arms $i, j$. For the bound of the Oracle queries up to $T$ for each arm, we have the following lemma.

**Lemma 5 (Oracle Queries Bound for Each Arm)** *For $(i,j) \in [d] \times [d]$, we always have*

$$\tau_{i,j}(T) = O(\log(Tm)).$$

*Proof.* For $i, j \in [d] \times [d]$, if $\tau_{i,j}$ is not the last stage for $i, j$, it holds that $|\mathcal{T}_{i,j}(\tau_{i,j})| \geq 2^{\tau_{i,j}-1}$. This can be derived from the update condition in the algorithm so that $|\mathcal{T}_{i,j}(\tau_{i,j})| \geq 2|\mathcal{T}_{i,j}(\tau_{i,j}-1)|$. Let $\tau_0 = \log(Tm)$. If $\tau \geq \tau_0 + 1$, for any $i, j \in [d] \times [d]$, we have

$$|\mathcal{T}_{i,j}(\tau)| \geq 2^{\tau-1} \geq 2^{\log(Tm)} = Tm.$$

Therefore, given the fact that the total number of selected bases over $T$ is at most $mT$, if $\tau_0 + 1 \leq \tau_{i,j}(T)$, there is always at most 1 pair of $((i,j),\tau)$ for the fixed $(i,j) \in [d] \times [d]$ and for all $\tau \in [\tau_0 + 1, \tau_{i,j}(T)]$. This implies that for $(i,j) \in [d] \times [d]$ s.t. $\tau_{i,j}(T) \geq \tau_0 + 1$, we have $\tau_{i,j}(T) \leq \tau_0 + 2$, which concludes the proof. ∎

From the above lemma, we can show that the oracle adaptivity complexity and query complexity are bounded by

$$\sum_{(i,j)\in[d]^2} \tau_{i,j}(T) = O(d^2 \log(Tm)).$$

Let $G_t = \sum_{s=1}^{t-1} D_{a_s} \Sigma D_{a_s} + D_\Sigma N_t + I$. For the regret bound, we first provide lemmas to define a favorable event of concentration bounds.

**Lemma 6 (Proposition 1 in [32])** *Let* $t \geq d(d+1)\log^3(T)/2$. *With probability at least* $1 - 1/(t\log(t))^2$, *for all* $a \in \mathcal{A}$,

$$|\langle a, \hat{\mu}_t - \mu \rangle| \leq f_t \|D_{n_t}^{-1} a\|_{G_t}.$$

From lemma 6, we define event $\mathcal{E}_{t,1} = \{|\langle a, \hat{\mu}_s - \mu \rangle| \leq f_s \|D_{n_s}^{-1} a\|_{Z_s} \; \forall s \in [\lceil d(d+1)\log^3(T)/2\rceil, t] \; \forall a \in \mathcal{A}\}$, which holds with probability of at least $1 - 1/t\log^2(t)$.

**Lemma 7 (Proposition 5 in [32])** *Let* $t \geq d(d+1)\log^3(T)/2$. *With probability at least* $1 - 1/(t\log(t))^2$, *for all* $(i,j) \in [d] \times [d]$, *we have*

$$|\hat{\Sigma}_{t,(i,j)} - \Sigma_{i,j}| \leq \frac{1}{4}\left(\frac{5h_t}{\sqrt{n_{t,(i,j)}}} + \frac{h_t^2}{n_{t,(i,j)}} + \frac{1}{n_{t,(i,j)}^2}\right).$$

From the above lemma, we define event $\mathcal{E}_{t,2} = \{|\hat{\Sigma}_{s,(i,j)} - \Sigma_{i,j}| \leq \frac{1}{4}(\frac{5h_s}{\sqrt{n_{s,(i,j)}}} + \frac{h_s^2}{n_{s,(i,j)}} + \frac{1}{n_{s,(i,j)}^2}), \forall s \in [\lceil d(d+1)\log^3(T)/2\rceil, t] \; \forall (i,j) \in [d]^2\}$, which holds with probability of at least $1 - 1/t\log^2(t)$.

Recall $\overline{\Sigma}_{t,(i,j)} = \hat{\Sigma}_{t,(i,j)} + \frac{1}{4}\left(\frac{5h_t}{\sqrt{n_{t,(i,j)}}} + \frac{h_t^2}{n_{t,(i,j)}} + \frac{1}{n_{t,(i,j)}^2}\right)$. Under $\mathcal{E}_{t,2}$, we have $\overline{\Sigma}_{t,(i,j)} \geq \Sigma_{i,j}$ for all $(i,j) \in [d]^2$. This implies that $\overline{G}_t \succeq G_t (\succeq 0)$ so that $\|D_{n_t}^{-1} a\|_{G_t} \leq \|D_{n_t}^{-1} a\|_{\overline{G}_t}$. Therefore, under $\mathcal{E}_t := \mathcal{E}_{t,1} \cap \mathcal{E}_{t,2}$, we have $|\langle a, \hat{\mu}_t - \mu \rangle| \leq f_t \|D_{n_t}^{-1} a\|_{\overline{G}_t}$, which implies

$$r_t^{UCB}(a) \geq \bar{r}(a).$$

Under $\mathcal{E}_t$, we can show that

$$
\begin{aligned}
\|D_{n_t}^{-1} a\|_{\overline{G}_t}^2 &= a^\top D_{n_t}^{-1} \overline{G}_t D_{n_t}^{-1} a \\
&= \sum_{(i,j)\in a\times a} \frac{\overline{G}_{t,(i,j)}}{n_{t,i} n_{t,j}} \\
&\lesssim \sum_{(i,j)\in a\times a} \frac{n_{t,(i,j)}\Sigma_{i,j}}{n_{t,i} n_{t,j}} + \sum_{(i,j)\in a\times a} \frac{h_t^2}{n_{t,(i,j)}^2} + \frac{h_t}{n_{t,(i,j)}^{3/2}} + \frac{1}{n_{t,(i,j)}^3} \\
&\leq \sum_{i\in a}\sum_{j\in a} \frac{n_{t,j}\Sigma_{i,j}}{n_{t,i} n_{t,j}} + \sum_{(i,j)\in a\times a} \frac{h_t^2}{n_{t,(i,j)}^2} + \frac{h_t}{n_{t,(i,j)}^{3/2}} + \frac{1}{n_{t,(i,j)}^3} \\
&\leq \sum_{i\in a} \frac{\sigma_i^2(a)}{n_{t,i}} + \sum_{(i,j)\in a\times a} \frac{h_t^2}{n_{t,(i,j)}^2} + \frac{h_t}{n_{t,(i,j)}^{3/2}} + \frac{1}{n_{t,(i,j)}^3},
\end{aligned}
$$

where the first inequality is obtained from $\hat{\Sigma}_{t,(i,j)} \leq \Sigma_{i,j} + \frac{1}{4}\left(\frac{5h_t}{\sqrt{n_{t,(i,j)}}} + \frac{h_t^2}{n_{t,(i,j)}} + \frac{1}{n_{t,(i,j)}^2}\right)$ under $\mathcal{E}_{t,2}$ and the second inequality is obtained from $n_{t,(i,j)} \leq n_{t,j}$.

For ease of presentation, we define auxiliary variables $n'_{t,(i,j)}$, $\hat{\mu}'_{t,(i)}$, $\overline{G}'_t$, and $f'_t$ for each $i \in [d]$ and time step $t \in [T]$ as follows: If the selected action at time $t$ is newly updated in the algorithm (Update = True), then

$$n'_{t,(i,j)} = n_{t,(i,j)}, \qquad \hat{\mu}'_{t,i} = \hat{\mu}_{t,i}, \qquad \overline{G}'_t = \overline{G}_t, \qquad f'_t = f_t.$$

Otherwise, if the previously selected action is maintained (Update = False), then

$$n'_{t,i} = n'_{t-1,i}, \qquad \hat{\mu}'_{t,i} = \hat{\mu}'_{t-1,i}, \qquad \overline{G}'_t = \overline{G}'_{t-1} \qquad f'_t = f'_{t-1}.$$

Using these adjusted statistics, we define the UCB-based optimistic reward estimate for any action $a$ as

$$\widetilde{r}^{\mathrm{UCB}}_t(a) = \langle a, \hat{\mu}'_t \rangle + f'_t \| D^{-1}_{n'_t} a \|_{\overline{G}'_t}.$$

Now we provide a bound for the regret as follows:

$$\mathcal{R}(T)$$

$$= \mathbb{E}\left[ \sum_{t \in [d(d+1)\log^3(T)/2, T]} (\bar{r}(a^*) - \bar{r}(a_t)) \mathbb{1}(\mathcal{E}_t) \right]$$

$$+ \mathbb{E}\left[ \sum_{t \in [d(d+1)\log^3(T)/2, T]} (\bar{r}(a^*) - \bar{r}(a_t)) \mathbb{1}(\mathcal{E}^c_t) \right] + \widetilde{O}(d^2)$$

$$\leq \mathbb{E}\left[ \sum_{t \in [d(d+1)\log^3(T)/2, T]} (\bar{r}(a^*) - \bar{r}(a_t)) \mathbb{1}(\mathcal{E}_t) \right] + \widetilde{O}(d^2)$$

$$\lesssim \mathbb{E}\left[ \sum_{t \in [d(d+1)\log^3(T)/2, T]} (\widetilde{r}^{UCB}_t(a^*) - \bar{r}(a_t)) \mathbb{1}(\mathcal{E}_t) \right] + \widetilde{O}(d^2)$$

$$\leq \mathbb{E}\left[ \sum_{t \in [d(d+1)\log^3(T)/2, T]} (\widetilde{r}^{UCB}_t(a_t) - \bar{r}(a_t)) \mathbb{1}(\mathcal{E}_t) \right] + \widetilde{O}(d^2)$$

$$\lesssim \mathbb{E}\left[ f_T \sum_{t \in [d(d+1)\log^3(T)/2, T]} \| D^{-1}_{n'_t} a_t \|_{\overline{G}'_t} \right]$$

$$\lesssim \mathbb{E}\left[ f_T \sqrt{ T \sum_{t \in [T]} \left( \sum_{i \in a_t} \frac{\sigma^2_i(a_t)}{n'_{t,i}} + \sum_{(i,j) \in a_t \times a_t} \frac{h^2_t}{n'^2_{t,(i,j)}} + \frac{h_t}{n'^{3/2}_{t,(i,j)}} + \frac{1}{n'^3_{t,(i,j)}} \right) } \right]$$

$$\lesssim \mathbb{E}\left[ f_T h_T \sqrt{ T \left( \sum_{i \in [d]} \sum_{\tau \in [\tau_{i,i}(T)]} \sum_{t \in \mathcal{T}_{i,i}(\tau)} \max_{a \in \mathcal{A}: i \in a} \frac{\sigma^2_i(a)}{n'_{t,i}} + \sum_{(i,j) \in [d]^2} \sum_{\tau \in [\tau_{i,j}(T)]} \sum_{t \in \mathcal{T}_{i,j}(\tau)} \frac{1}{n'^{3/2}_{t,(i,j)}} \right) } \right].$$

For bounding the last term, we have

$$
\mathbb{E}\left[f_T h_T \sqrt{T\left(\sum_{i\in[d]}\sum_{\tau\in[\tau_{i,i}(T)]}\sum_{t\in\mathcal{T}_{i,i}(\tau)}\max_{a\in\mathcal{A}:i\in a}\frac{\sigma_i^2(a_t)}{n'_{t,i}}+\sum_{(i,j)\in[d]^2}\sum_{\tau\in[\tau_{i,j}(T)]}\sum_{t\in\mathcal{T}_{i,j}(\tau)}\frac{1}{n'^{3/2}_{t,(i,j)}}\right)}\right]
$$

$$
\leq\mathbb{E}\left[f_T h_T\sqrt{T\left(\sum_{i\in[d]}\sum_{\tau\in[\tau_{i,i}(T)]}\sum_{t\in\mathcal{T}_{i,i}(\tau)}\max_{a\in\mathcal{A}:i\in a}\frac{2\sigma_i^2(a)}{|\mathcal{T}_{i,i}(\tau-1)|}+\sum_{(i,j)\in[d]^2}\sum_{\tau\in[\tau_{i,j}(T)]}\sum_{t\in\mathcal{T}_{i,j}(\tau)}\frac{1}{|\mathcal{T}_{i,j}(\tau-1)|^{3/2}}\right)}\right]
$$

$$
\leq\mathbb{E}\left[f_T h_T\sqrt{T\left(\sum_{i\in[d]}\sum_{\tau\in[\tau_{i,i}(T)]}|\mathcal{T}_{i,i}(\tau)|\max_{a\in\mathcal{A}:i\in a}\frac{2\sigma_i^2(a)}{|\mathcal{T}_{i,i}(\tau-1)|}+\sum_{(i,j)\in[d]^2}\sum_{\tau\in[\tau_{i,j}(T)]}|\mathcal{T}_{i,j}(\tau)|\frac{1}{|\mathcal{T}_{i,j}(\tau-1)|^{3/2}}\right)}\right]
$$

$$
\leq\mathbb{E}\left[f_T h_T\sqrt{T\left(\sum_{i\in[d]}\sum_{\tau\in[\tau_{i,i}(T)]}4|\mathcal{T}_{i,i}(\tau-1)|\frac{\max_{a\in\mathcal{A}:i\in a}\sigma_i^2(a)}{|\mathcal{T}_i(\tau-1)|}+\sum_{(i,j)\in[d]^2}\sum_{\tau\in[\tau_{i,j}(T)]}2|\mathcal{T}_{i,j}(\tau-1)|\frac{1}{|\mathcal{T}_{i,j}(\tau-1)|^{3/2}}\right)}\right]
$$

$$
\lesssim\mathbb{E}\left[f_T h_T\sqrt{T\left(\sum_{i\in[d]}\tau_{i,i}(T)\max_{a\in\mathcal{A}}\sigma_i^2(a)+\sum_{(i,j)\in[d]^2}\tau_{i,j}(T)\frac{1}{\sqrt{|\mathcal{T}_{i,j}(\tau-1)|}}\right)}\right]
$$

$$
\lesssim f_T h_T\sqrt{T\sum_{i\in[d]}\max_{a\in\mathcal{A}}\sigma_i^2(a)\log(Tm)},
$$

where the first inequality is obtained from $n'_{t,i}\geq|\mathcal{T}_i(\tau-1)|$, the third inequality is obtained from the condition of updates in the algorithm, and the last inequality is obtained from Lemma 5, $|\mathcal{T}_{i,j}(\tau-1)|\geq\log^3(T)$ from warm-up stage in the algorithm, and large enough $T$. This concludes the proof with the fact that $f_T=O(\log(T))$ when $T$ is large enough.

**Oracle Complexity Bounds.** Based on Lemma 5, we can show that the oracle query complexity is bounded by $O(d^2\log(Tm))$. Since the epochs for each base arm are updated separately, the adaptivity complexity is also bounded by $O(d^2\log(Tm))$.

### A.6 Proof of Theorem 4

From lemma 6, we define event $\mathcal{E}_1=\{|\langle a,\hat{\mu}_\tau-\mu\rangle|\leq f_T\|D^{-1}_{n_\tau}a\|_{Z_\tau}\forall a\in\mathcal{A},\forall\tau\in[M]\}$, which holds with probability of at least $1-1/(T\log^2(T))$. From Lemma 7, we define event $\mathcal{E}_2=\{|\hat{\Sigma}_{\tau,(i,j)}-\Sigma_{i,j}|\leq\frac{1}{4}(\frac{5h_T}{\sqrt{n_{\tau,(i,j)}}}+\frac{h_T^2}{n_{\tau,(i,j)}}+\frac{1}{n^2_{\tau,(i,j)}}),\forall(i,j)\in[d]^2,\forall\tau\in[M]\}$, which holds with probability of at least $1-1/(T\log^2(T))$.

Under $\mathcal{E}_2$, we have $\overline{\Sigma}_{\tau,(i,j)}\geq\Sigma_{i,j}$ for all $(i,j)\in[d]^2$. This implies that $\overline{G}_\tau\succeq G_\tau(\succeq 0)$ so that $\|D^{-1}_{n_\tau}a\|_{G_\tau}\leq\|D^{-1}_{n_\tau}a\|_{\overline{G}_\tau}$. Therefore, under $\mathcal{E}:=\mathcal{E}_1\cap\mathcal{E}_2$, we have $|\langle a,\hat{\mu}_\tau-\mu\rangle|\leq f_\tau\|D^{-1}_{n_\tau}a\|_{\overline{G}_\tau}$, which implies

$$
r_\tau^{UCB}(a)\geq\bar{r}(a)\geq r_\tau^{LCB}(a).
$$

Similar to Lemma 4, we then have the following lemma.

**Lemma 8** *Under $\mathcal{E}$, we can show that for all $\tau\in[M]$, $a^*\in\mathcal{A}_\tau$.*

*Proof.* This can be shown by induction. Suppose $a^*\in\mathcal{A}_\tau$ for fixed $\tau\in[M-1]$. Under $\mathcal{E}$, we have $\bar{r}(a)\leq r_{\tau+1}^{UCB}(a)$ and $\bar{r}(a)\geq r_{\tau+1}^{LCB}(a)$ for any $a\in\mathcal{A}_\tau$. Then for any fixed $i\in a^*$, for any $a\in\mathcal{A}_\tau$, we can show that

$$
r_{\tau+1}^{UCB}(a_{\tau+1}^{(i)})\geq r_{\tau+1}^{UCB}(a^*)\geq\bar{r}(a^*)\geq r^{LCB}(a),
$$

where the first inequality is obtained from the definition of $a_{\tau+1}^{(i)}$ with $a^*\in\mathcal{A}_\tau$. This implies that

$$
r_{\tau+1}^{UCB}(a_{\tau+1}^{(i)})\geq\max_{a\in\mathcal{A}_\tau}r_{\tau+1}^{LCB}(a),
$$

which implies that $i \in a^*$ is not eliminated from the elimination condition at the $\tau + 1$-th epoch. This holds for all $i \in a^*$ so that $a^* \in \mathcal{A}'_{\tau+1}$.

Then for any fixed $i \in a^*$ and $j \in a^*/\{i\}$, for any $a \in \mathcal{A}'_{\tau+1}$, we can show that

$$r_{\tau+1}^{UCB}(a_{\tau+1}^{(i,j)}) \geq r_{\tau+1}^{UCB}(a^*) \geq \bar{r}(a^*) \geq r^{LCB}(a),$$

where the first inequality is obtained from the definition of $a_{\tau+1}^{(i,j)}$ with $a^* \in \mathcal{A}'_{\tau+1}$. This implies that

$$r_{\tau+1}^{UCB}(a_{\tau+1}^{(i,j)}) \geq \max_{a \in \mathcal{A}'_{\tau+1}} r_{\tau+1}^{LCB}(a),$$

which implies that $i \in a^*$ and $j \in a^*/\{i\}$ are not eliminated from the elimination condition at the $\tau + 1$-th epoch. This holds for all $i \in a^*$ and $j \in a^*/\{i\}$ so that $a^* \in \mathcal{A}_{\tau+1}$. With $a^* \in \mathcal{A}_0 = \mathcal{A}$, we can conclude the induction. ∎

Under $\mathcal{E}$, we can show that

$$\|D_{n_\tau}^{-1} a\|_{\overline{G}_\tau}^2 = a^\top D_{n_\tau}^{-1} \overline{G}_\tau D_{n_\tau}^{-1} a$$

$$= \sum_{(i,j) \in a \times a} \frac{\overline{G}_{\tau,(i,j)}}{n_{\tau,(i,i)} n_{\tau,(j,j)}}$$

$$\lesssim \sum_{(i,j) \in a \times a} \frac{n_{\tau,(i,j)} \Sigma_{i,j}}{n_{\tau,(i,i)} n_{\tau,(j,j)}} + \sum_{(i,j) \in a \times a} \frac{h_T^2}{n_{\tau,(i,j)}^2} + \frac{h_T}{n_{\tau,(i,j)}^{3/2}} + \frac{1}{n_{\tau,(i,j)}^3}$$

$$\leq \sum_{i \in a} \sum_{j \in a} \frac{n_{\tau,(j,j)} \Sigma_{i,j}}{n_{\tau,(i,i)} n_{\tau,(j,j)}} + \sum_{(i,j) \in a \times a} \frac{h_T^2}{n_{\tau,(i,j)}^2} + \frac{h_T}{n_{\tau,(i,j)}^{3/2}} + \frac{1}{n_{\tau,(i,j)}^3}$$

$$\leq \sum_{i \in a} \frac{\sigma_i^2(a)}{n_{\tau,(i,i)}} + \sum_{(i,j) \in a \times a} \frac{h_T^2}{n_{\tau,(i,j)}^2} + \frac{h_T}{n_{\tau,(i,j)}^{3/2}} + \frac{1}{n_{\tau,(i,j)}^3}, \tag{5}$$

where the first inequality is obtained from $\hat{\Sigma}_{\tau,(i,j)} \leq \Sigma_{i,j} + \frac{1}{4}\left(\frac{5h_T}{\sqrt{n_{\tau,(i,j)}}} + \frac{h_T^2}{n_{\tau,(i,j)}} + \frac{1}{n_{\tau,(i,j)}^2}\right)$ under $\mathcal{E}_2$ and the second inequality is obtained from $n_{\tau,(i,j)} \leq n_{\tau,(j,j)}$.

Then, under $\mathcal{E}$, for $t \in \mathcal{T}_\tau^{(1)} \cup \mathcal{T}_\tau^{(2)}$ we have

$$\bar{r}(a^*) - \bar{r}(a_t) \lesssim r_\tau^{LCB}(a^*) + f_T \sqrt{\sum_{i \in a^*} \frac{\sigma_i^2(a^*)}{n_{\tau,(i,i)}} + \sum_{(i,j) \in a^* \times a^*} \frac{h_T^2}{n_{\tau,(i,j)}^2} + \frac{h_T}{n_{\tau,(i,j)}^{3/2}} + \frac{1}{n_{\tau,(i,j)}^3}}$$

$$- r_\tau^{UCB}(a_t) + f_T \sqrt{\sum_{i \in a_t} \frac{\sigma_i^2(a_t)}{n_{\tau,(i,i)}} + \sum_{(i,j) \in a_t \times a_t} \frac{h_T^2}{n_{\tau,(i,j)}^2} + \frac{h_T}{n_{\tau,(i,j)}^{3/2}} + \frac{1}{n_{\tau,(i,j)}^3}}$$

$$\lesssim f_T \sqrt{\max_{a \in \mathcal{A}_\tau} \left(\sum_{i \in a} \frac{\sigma_i^2(a)}{n_{\tau,(i,i)}} + \sum_{(i,j) \in a \times a} \frac{h_T^2}{n_{\tau,(i,j)}^{3/2}}\right)}$$

$$\lesssim f_T \sqrt{\max_{a \in \mathcal{A}_\tau} \left(\sum_{i \in a} \frac{d\sigma_i^2(a)}{T_{\tau-1} - (dm^2 T_{\tau-1} \log(T))^{2/3}} + \frac{dh_T^2}{T_{\tau-1} \log(T)}\right)}$$

$$\lesssim f_T h_T \sqrt{d \max_{a \in \mathcal{A}_\tau} \sum_{i \in a} \frac{\sigma_i^2(a)}{T_{\tau-1}}},$$

where the first inequality is obtained from $\mathcal{E}$, the second inequality comes from the fact that $a^* \in \mathcal{A}_{\tau-1}$ from Lemma 8 and elimination conditions from the algorithm, and the third inequality is obtained from $n_{\tau,(i,i)} \geq \sum_{t \in \mathcal{T}_{\tau-1}^{(1)}} \mathbb{1}(i \in a_t) \gtrsim (T_{\tau-1} - (dm^2 T_{\tau-1} \log(T))^{2/3})/d$ and $n_{\tau,(i,j)} \geq \sum_{t \in \mathcal{T}_{\tau-1}^{(2)}} \mathbb{1}(i \in a_t)\mathbb{1}(j \in a_t) \gtrsim (dm^2 T_{\tau-1} \log(T))^{2/3}/d^2$ for $i \neq j$ from the exploration in the algorithm at the $\tau - 1$-th epoch, and the last inequality is obtained from large enough $T$.

Finally, we can show that

$$\mathcal{R}(T) \leq \mathbb{E}\left[\sum_{\tau=1}^{M}\sum_{t\in\mathcal{T}_\tau}\bar{r}(a^*)-\bar{r}(a_t)\mid\mathcal{E}\right]\mathbb{P}(\mathcal{E}) + \mathbb{E}\left[\sum_{\tau=1}^{M}\sum_{t\in\mathcal{T}_\tau}\bar{r}(a^*)-\bar{r}(a_t)\mid\mathcal{E}^c\right]\mathbb{P}(\mathcal{E}^c) + O(d^2)$$

$$\lesssim \mathbb{E}\left[\sum_{\tau=1}^{M}\sum_{t\in\mathcal{T}_\tau}\bar{r}(a^*)-\bar{r}(a_t)\mid\mathcal{E}\right] + \frac{md}{T} + O(d^2)$$

$$\lesssim \sum_{\tau=1}^{M} T_\tau f_T h_T \sqrt{d\left(\max_{a\in\mathcal{A}}\left(\sum_{i\in a}\frac{\sigma_i^2(a)}{T_{\tau-1}}\right) + \frac{1}{T_{\tau-1}}\right)}$$

$$\leq \sum_{\tau=1}^{M} \eta f_T h_T \sqrt{d\max_{a\in\mathcal{A}}\sum_{i\in a}\sigma_i^2(a)}$$

$$\leq \eta M f_T h_T \sqrt{d\max_{a\in\mathcal{A}}\sum_{i\in a}\sigma_i^2(a)}$$

$$\lesssim f_T h_T \log\log(T)\sqrt{d\max_{a\in\mathcal{A}}\sum_{i\in a}\sigma_i^2(a)T},$$

which concludes the proof with the fact that $f_T = O(\log(T))$ when $T$ is large enough.

**Oracle Complexity Bounds.** Based on the oracle calls in Lines 15,16, 17, and 19 of Algorithm 4, we observe that each epoch involves at most $d + d^2$ independent oracle queries (Lines 15,16) and two sequential oracle queries (Lines 17,19). Since the total number of epochs is $M = \Theta(\log\log T)$, the overall adaptivity complexity is bounded by $\Theta(\log\log T)$, and the query complexity is bounded by $O(d^2 \log\log T)$.

### A.7 Rare Oracle Queries for General-Reward CMAB

#### A.7.1 Adaptive Rare Oracle Queries for General Reward CMAB

We first propose an algorithm (Algorithm 6) for rare oracle queries for general reward CMAB using the adaptive framework as in Algorithm 1. For $i \in [d]$, let $\hat{F}_{\tau,i}(x)$ be the fraction of the observed feedback from arm $i$ that is no longer than $0 \leq x \leq 1$ before time $t$. By inspired by [4], for $i \in [d]$, we define $\underline{\mathcal{D}}_{t,i}$ to be the distribution whose CDF is, for some constant $C > 0$,

$$\underline{F}_{t,i}(x) = \begin{cases} \max\{\hat{F}_{t,i}(x) - \sqrt{\frac{C\ln t}{n_{t,i}}}, 0\} & \text{if } 0 \leq x < 1 \\ 1 & \text{if } x = 1. \end{cases}$$

Then we construct UCB for each action based on $\underline{\mathcal{D}}_t$ as follows:

$$r_t^{UCB}(a) = \mathbb{E}_{x\sim\underline{\mathcal{D}}_t}[r(x,a)] \tag{6}$$

#### A.7.2 Scheduled Rare Oracle Queries for General Reward CMAB

For $i \in [d]$, let $\hat{F}_{\tau,i}(x)$ be the fraction of the observed feedback from arm $i$ that is no longer than $0 \leq x \leq 1$ before epoch $\tau$. Then, for some constant $C > 0$, we define $\underline{\mathcal{D}}_{\tau,i}$ to be the distribution whose CDF is

$$\underline{F}_{\tau,i}(x) = \begin{cases} \max\{\hat{F}_{\tau,i}(x) - \sqrt{\frac{C\ln T}{n_{\tau,i}}}, 0\} & \text{if } 0 \leq x < 1 \\ 1 & \text{if } x = 1, \end{cases}$$

and define $\overline{\mathcal{D}}_{\tau,i}$ to be the distribution whose CDF is

$$\overline{F}_{\tau,i}(x) = \begin{cases} \min\{\hat{F}_{\tau,i}(x) + \sqrt{\frac{C\ln T}{n_{\tau,i}}}, 1\} & \text{if } 0 \leq x < 1 \\ 1 & \text{if } x = 1. \end{cases}$$

**Algorithm 6** Adaptive Rare Oracle Queries for General-Reward CMAB (`AROQ-GR-CMAB`)

**Init:** $\tau_i = 1$ for all $i \in [d]$
**for** $t = 1, 2..., T$ **do**

 **for** $i \in [d]$ *s.t.* $|\mathcal{T}_i(\tau_i)| \geq 1 + \sqrt{Tm \cdot |\mathcal{T}_i(\tau_i - 1)|/d}$ **do**
  $\tau_i \leftarrow \tau_i + 1, \mathcal{T}_i(\tau_i) \leftarrow \emptyset$
  $Update \leftarrow True$
 **if** $Update = True$ **then**
  $a_t \leftarrow \arg\max_{a \in \mathcal{A}} r_t^{UCB}(a)$ with (6)        *// Oracle Query*
  $Update \leftarrow False$
 **else**
  $a_t \leftarrow a_{t-1}$
 Play $a_t$ and observe feedback $y_{t,i}$ for $i \in a_t$
 $\mathcal{T}_i(\tau_i) \leftarrow \mathcal{T}_i(\tau_i) \cup \{t\}$ for all $i \in a_t$

We construct UCB and LCB for each action based on $\underline{\mathcal{D}}_\tau$ and $\overline{\mathcal{D}}_\tau$, respectively, as follows:

$$r_\tau^{UCB}(a) = \mathbb{E}_{x \sim \underline{\mathcal{D}}_\tau} = [\bar{r}(x,a)] \text{ and } r_\tau^{LCB}(a) = \mathbb{E}_{x \sim \overline{\mathcal{D}}_\tau} = [\bar{r}(x,a)] \tag{7}$$

Let grid $\mathcal{T} = \{t_1(=1), \ldots, t_M(=T)\}$, where $t_\tau = \eta\sqrt{t_{\tau-1}}$ and $\eta = T^{\frac{1}{2-2^{1-M}}}$ for $M > 0$. We set $M = \Theta(\log\log(T))$.

---

**Algorithm 7** Scheduled Rare Oracle Queries for General-Reward CMAB (`SROQ-GR-CMAB`)

 **Input:** $\mathcal{T}$
30 **for** $\tau = 1, 2, \ldots, M$ **do**
31  Update $\hat{F}_\tau(x)$
32  $a_\tau^{(i)} := \arg\max_{a \in \mathcal{A}_{\tau-1}: i \in a} r_\tau^{UCB}(a)$ for all $i \in \mathcal{N}_{\tau-1}$ with (7)   *// Oracle Queries*
33  $\mathcal{N}_\tau \leftarrow \{i \in \mathcal{N}_{\tau-1} \mid r_\tau^{UCB}(a_\tau^{(i)}) \geq \max_{a \in \mathcal{A}_{\tau-1}} r_\tau^{LCB}(a)\}$ with (7)   *// Oracle Query*
34  $\mathcal{A}_\tau \leftarrow \{a \in \mathcal{A}_{\tau-1} \mid a_i = 0 \text{ for all } i \in [d]/\mathcal{N}_\tau\}$
35  $\mathcal{T}_\tau \leftarrow [t_\tau, t_{\tau+1} - 1]$
36  **for** $t \in \mathcal{T}_\tau$ **do**
37   $i \leftarrow \big((t-1) \bmod |\mathcal{N}_\tau| + 1\big)$-th element in $\mathcal{N}_\tau$
38   Play $a_t = a_\tau^{(i)}$ and observe feedback $y_{t,i}$ for $i \in a_t$

---

### A.8 Proof of Theorem 5

Let $\tau_i(t)$ be the value of $\tau_i$ at time $t$ in the algorithm. Then $\tau_i(T)$ represents the number of updates, each update requiring Oracle queries, up to $T$ from arm $i$. For a slight abuse of notation, we use $\mathcal{T}_i(\tau_i)$ for the set $\mathcal{T}_i(\tau_i)$ in the algorithm at the last time step $T$. For the bound of the total Oracle queries up to $T$, from Lemma 1, we have

$$\mathbb{E}\left[\sum_{i \in [d]} \tau_i(T)\right] = O(d \log\log(Tm/d)). \tag{8}$$

For ease of presentation, we use $\bar{r}_{\mathcal{D}}(a) = \mathbb{E}_{X \sim \mathcal{D}}[r(a, X)]$.

**Lemma 9 (Lemma 3 in [4])** *Let* $\mathbb{P} = \mathbb{P}_1 \times \cdots \times \mathbb{P}_d$ *and* $\mathbb{P}' = \mathbb{P}'_1 \times \cdots \times \mathbb{P}'_d$ *be two probability distributions* $\mathcal{D}$ *and* $\mathcal{D}'$, *respectively, over* $[0, 1]^d$. *Let* $F_i$ *and* $F'_i$ *be the CDFs of* $\mathbb{P}_i$ *and* $\mathbb{P}'_i$, *respectively for* $i \in [d]$. *Suppose each* $\mathbb{P}_i$ *is a discrete distribution with finite support.*

 *(a) If we have* $F'_i(x) \leq F_i(x)$ *for any* $i \in [d], x \in [0, 1]$, *then for any* $a \in \mathcal{A}$, *we have* $\bar{r}_{\mathcal{D}'}(a) \geq \bar{r}_{\mathcal{D}}(a)$.

(b) If we have $F_i(x) - F'_i(x) \leq z_i$ with $z_i > 0$ for any $i \in [d]$, $x \in [0, 1]$, then for any $a \in \mathcal{A}$, we have $\bar{r}_{\mathcal{D}'}(a) - \bar{r}_{\mathcal{D}}(a) \leq 2L \sum_{i \in a} z_i$.

**Lemma 10 (Dvoretzky-Kiefer-Wolfowitz inequality)** *For i.i.d. samples of $X_1, \ldots, X_n$ drawn from a distribution $\mathcal{D}$, let empirical CDF $\hat{F}_n(x) = \frac{1}{n} \sum_{i=1}^{n} \mathbb{1}(X_i \leq x)$. Then, for any $\epsilon > 0$ and any $n \in \mathbb{N}$, we have*

$$\mathbb{P}\left[\sup_{x \in \mathbb{R}} |\hat{F}_n(x) - F(x)| \geq \epsilon\right] \leq 2 \exp^{-2n\epsilon^2}.$$

From the above lemma, we define favorable event $\mathcal{E}_t = \{\sup_{x \in [0,1]} |\hat{F}_{i,n_{s,i}}(x) - F_i(x)| \leq \sqrt{\frac{3 \ln s}{2n_{s,i}}} \, \forall s \in [1, t] \, \forall i \in [d]\}$, which holds with probability at least $1 - O(d/t^2)$. Recall that $r_t^{UCB}(a) = \mathbb{E}_{y \sim \underline{\mathcal{D}}_t}[\bar{r}(y, a)]$.

For ease of presentation, we define auxiliary variables $\underline{\mathcal{D}}'_t$ and $n'_{t,i}$ for $i \in [d]$ and time step $t \in [T]$ as follows: If the selected action at time $t$ is newly updated in the algorithm (Update = True), then $\underline{\mathcal{D}}'_t = \underline{\mathcal{D}}_t$ and $n'_{t,i} = n_{t,i}$. Otherwise, if the previously selected action is maintained (Update = False), then $\underline{\mathcal{D}}'_t = \underline{\mathcal{D}}'_{t-1}$ and $n'_{t,i} = n'_{t-1,i}$. Using these adjusted statistics, we define the UCB-based optimistic reward estimate for any action $a$ as

$$\tilde{r}_t^{UCB}(a) = \mathbb{E}_{x \sim \underline{\mathcal{D}}'_t}[\bar{r}(x, a)].$$

Now we provide a bound for the regret as follows:

$$
\begin{aligned}
\mathcal{R}(T) &= \mathbb{E}\left[\sum_{t \in [T]} (\bar{r}(a^*) - \bar{r}(a_t)) \mathbb{1}(\mathcal{E}_t)\right] + \mathbb{E}\left[\sum_{t \in [T]} (\bar{r}(a^*) - \bar{r}(a_t)) \mathbb{1}(\mathcal{E}_t^c)\right] \\
&\leq \mathbb{E}\left[\sum_{t \in [T]} (\bar{r}(a^*) - \bar{r}(a_t)) \mathbb{1}(\mathcal{E}_t)\right] + L \sum_{t \in [T]} \sum_{l=1}^{t-1} \sum_{i \in [d]} \mathbb{P}\left(\sup_{x \in [0,1]} |\hat{F}_{i,l}(x) - F_i(x)| \geq \sqrt{\frac{3 \ln t}{2l}}\right) \\
&\lesssim \mathbb{E}\left[\sum_{t \in [T]} (\tilde{r}_t^{UCB}(a^*) - \bar{r}(a_t)) \mathbb{1}(\mathcal{E}_t)\right] + Ld \\
&\lesssim \mathbb{E}\left[\sum_{t \in [T]} (\tilde{r}_t^{UCB}(a_t) - \bar{r}(a_t)) \mathbb{1}(\mathcal{E}_t)\right] \\
&\lesssim \mathbb{E}\left[L \sum_{t \in [T]} \sum_{i \in a_t} \sqrt{\frac{\log T}{n'_{t,i}}}\right],
\end{aligned}
$$

where the second inequality is obtained from (a) in Lemma 9 and the last inequality comes from $\mathcal{E}_t$ and (b) in Lemma 9. For bounding the last term, we have

$$\mathbb{E}\left[L \sum_{t\in[T]} \sum_{i\in a_t} \sqrt{\frac{\log T}{n'_{t,i}}}\right]$$

$$= \mathbb{E}\left[L \sum_{i\in[d]} \sum_{\tau\in[\tau_i(T)]} \sum_{t\in\mathcal{T}_i(\tau)} \sqrt{\frac{\log T}{n'_{t,i}}}\right]$$

$$\leq \mathbb{E}\left[L \sum_{i\in[d]} \sum_{\tau\in[\tau_i(T)]} \sum_{t\in\mathcal{T}_i(\tau)} \sqrt{\frac{\log T}{|\mathcal{T}_i(\tau-1)|}}\right]$$

$$= \mathbb{E}\left[L \sum_{i\in[d]} \sum_{\tau\in[\tau_i(T)]} |\mathcal{T}_i(\tau)| \sqrt{\frac{\log T}{|\mathcal{T}_i(\tau-1)|}}\right]$$

$$= \mathbb{E}\left[L \sum_{i\in[d]} \sum_{\tau\in[\tau_i(T)]} \sqrt{\frac{Tm\cdot|\mathcal{T}_i(\tau-1)|}{d}} \sqrt{\frac{\log T}{|\mathcal{T}_i(\tau-1)|}}\right]$$

$$= \mathbb{E}\left[L \sum_{i\in[d]} \tau_i(T)\right] \sqrt{\frac{Tm\log T}{d}}$$

$$\lesssim L \log\log(Tm/d) \sqrt{dmT\log T},$$

where the first inequality is obtained from $n'_{t,i} \geq |\mathcal{T}_i(\tau-1)|$ for $t \in \mathcal{T}_i(\tau)$, the second equality is obtained from the condition of updates in the algorithm, and the last inequality is obtained from (8).

**Oracle Complexity Bounds.** Based on (8), we can show that the oracle query complexity is bounded by $O(d\log\log(Tm/d))$. Since the epochs for each base arm are updated separately, the adaptivity complexity is also bounded by $O(d\log\log(Tm/d))$.

## A.9 Proof of Theorem 6

From Lemma 10, we define the event $\mathcal{E} = \{\sup_{x\in[0,1]} |\hat{F}_{\tau,i}(x) - F_i(x)| \leq \sqrt{\frac{3\ln T}{2n_{\tau,i}}} \, \forall \tau \in [M], \forall i \in [d]\}$. Then, similar to Lemma 4, under the event $\mathcal{E}$, we show that $A_\tau$, activated arm set for $\tau$-th epoch, always contains the optimal arm $a^*$.

**Lemma 11** *Under $\mathcal{E}$, we can show that for all $\tau \in [M]$, $a^* \in \mathcal{A}_\tau$.*

*Proof.* The proof is the same as that of Lemma 4. This can be shown by induction. Suppose $a^* \in \mathcal{A}_\tau$ for fixed $\tau \in [M]$. Under $\mathcal{E}$, we have $\bar{r}(a) \leq r_{\tau+1}^{UCB}(a)$ and $\bar{r}(a) \geq r_{\tau+1}^{LCB}(a)$ for any $a \in \mathcal{A}_\tau$. Then for any fixed $i \in a^*$, for any $a \in \mathcal{A}_\tau$, we can show that

$$r_{\tau+1}^{UCB}(a_{\tau+1}^{(i)}) \geq r_{\tau+1}^{UCB}(a^*) \geq \bar{r}(a^*) \geq r^{LCB}(a),$$

where the first inequality is obtained from the definition of $a_{\tau+1}^{(i)}$ with $a^* \in \mathcal{A}_\tau$. This implies that

$$r_{\tau+1}^{UCB}(a_{\tau+1}^{(i)}) \geq \max_{a\in\mathcal{A}_\tau} r_{\tau+1}^{LCB}(a),$$

which implies that $i \in a^*$ is not eliminated from the elimination condition at the $\tau+1$-th epoch. This holds for all $i \in a^*$ so that $a^* \in \mathcal{A}_{\tau+1}$. With $a^* \in \mathcal{A}_0 = \mathcal{A}$, we can conclude the induction. $\blacksquare$

Under $\mathcal{E}$, we have

$$
\bar{r}(a^*) - \bar{r}(a_\tau^{(i)}) \le r_\tau^{LCB}(a^*) + 2L \sum_{j \in a^*} \sqrt{\frac{1.5 \log T}{n_{\tau,j}}} - r_\tau^{UCB}(a_\tau^{(i)}) + 2L \sum_{j \in a_\tau^{(i)}} \sqrt{\frac{1.5 \log T}{n_{\tau,j}}}
$$

$$
\lesssim L \sum_{j \in a^*} \sqrt{\frac{\log T}{n_{\tau,j}}} + L \sum_{j \in a_\tau^{(i)}} \sqrt{\frac{\log T}{n_{\tau,j}}}
$$

$$
\lesssim L \sum_{j \in a^*} \sqrt{\frac{d \log T}{|\mathcal{T}_{\tau-1}|}} + L \sum_{j \in a_\tau^{(i)}} \sqrt{\frac{d \log T}{|\mathcal{T}_{\tau-1}|}}
$$

$$
\lesssim Lm \sqrt{\frac{d \log T}{|\mathcal{T}_{\tau-1}|}},
$$

where the first inequality is obtained from $\mathcal{E}$, the second inequality comes from the fact that $a^* \in \mathcal{A}_{\tau-1}$ from Lemma 11, and $\max_{a \in \mathcal{A}_{\tau-1}} r_\tau^{LCB}(a) \le r_\tau^{UCB}(a_\tau^{(i)})$ from the algorithm, and the third inequality is obtained from $n_{\tau,i} \ge \sum_{t \in \mathcal{T}_{\tau-1}} \mathbb{1}(i \in a_t) \gtrsim |\mathcal{T}_{\tau-1}|/d$ from the exploration in the algorithm at the $\tau - 1$-th epoch.

Finally, we can show that

$$
\mathcal{R}(T) \le \mathbb{E}\left[\sum_{\tau=1}^M \sum_{t \in \mathcal{T}_\tau} \bar{r}(a^*) - \bar{r}(a_t) \mid \mathcal{E}\right] \mathbb{P}(\mathcal{E}) + \mathbb{E}\left[\sum_{\tau=1}^M \sum_{t \in \mathcal{T}_\tau} \bar{r}(a^*) - \bar{r}(a_t) \mid \mathcal{E}^c\right] \mathbb{P}(\mathcal{E}^c)
$$

$$
\lesssim \mathbb{E}\left[\sum_{\tau=1}^M \sum_{t \in \mathcal{T}_\tau} \bar{r}(a^*) - \bar{r}(a_t) \mid \mathcal{E}\right] + L \sum_{t \in [T]} \sum_{l=1}^{t-1} \sum_{i \in [d]} \mathbb{P}\left(\sup_{x \in [0,1]} |\hat{F}_{i,l}(x) - F_i(x)| \ge L\sqrt{\frac{3 \ln t}{2l}}\right)
$$

$$
\lesssim \sum_{\tau=1}^M |\mathcal{T}_\tau| Lm \sqrt{\frac{d \log T}{|\mathcal{T}_{\tau-1}|}}
$$

$$
\le \sum_{\tau=1}^M \eta Lm \sqrt{d |\mathcal{T}_{\tau-1}| \log T}
$$

$$
= Lm \eta \sqrt{dMT \log T}
$$

$$
\lesssim Lm \sqrt{dT \log(T) \log \log(T)},
$$

which concludes the proof.

**Oracle Complexity Bounds.** Based on the oracle calls in Lines 32 and 33 of Algorithm 7, we observe that each epoch involves at most $d$ independent oracle queries (Line 32) and one sequential oracle query (Line 33). Since the total number of epochs is $M = \Theta(\log \log T)$, the overall adaptivity complexity is bounded by $\Theta(\log \log T)$, and the query complexity is bounded by $O(d \log \log T)$.

### A.10 Extension to Continuous Distributions for General Reward Functions

We now consider the setting where each $y_{t,i} \in [0,1]$ is drawn from a continuous distribution $\mathcal{D}$. In this case, we impose the additional Lipschitz-continuity assumption on the reward function as follows.

**Assumption 2** *There exists $C > 0$ such that for any $a \in \mathcal{A}$ and any $y, y' \in [0,1]^m$, we have $|r(a,y) - r(a,y')| \le C \sum_{i \in a} |y_i - y_i'|$.*

Here we provide a regret bound for the discretization of Algorithm 6, and that of Algorithm 7 is omitted due to its redundancy.

**Theorem 8** *Algorithm 8 with Algorithm 6 achieves a regret bound of $\mathcal{R}(T) = \widetilde{O}(L\sqrt{mdT})$*

---

**Algorithm 8** Discretizations [4]

---

Set the number of intervals $s \leftarrow \lceil C\sqrt{mT} \rceil$
**for** $j = 1$ *to* $s$ **do**

  Define interval $I_j \leftarrow \begin{cases} [0, \frac{1}{s}], & \text{if } j = 1 \\ (\frac{j-1}{s}, \frac{j}{s}], & \text{if } j = 2, \ldots, s \end{cases}$

Invoke Algorithm 6 or Algorithm 7 for $T$ rounds with the following modification:
**for** $i \in a_t$ **do**

  Upon observing an outcome $y_{t,i} \in [0, 1]$, identify $j \in [s]$ such that $y_{t,i} \in I_j$
  Treat the observation as $\frac{j}{s}$

---

*Proof.* We define $\widetilde{\mathcal{D}}$ to be the discretized distribution of $\mathcal{D}$. For ease of presentation, we use $\bar{r}_{\mathcal{D}}(a) = \mathbb{E}_{X \sim \mathcal{D}}[r(a, X)]$.

**Lemma 12 (Lemma 7 in [4])** *For any $a \in \mathcal{A}$, we have $|\bar{r}_{\mathcal{D}}(a) - \bar{r}_{\widetilde{\mathcal{D}}}(a)| \leq \sqrt{\frac{m}{T}}$.*

Then, from the above lemma, we have

$$
\begin{aligned}
\mathcal{R}(T) = \mathbb{E}\left[\sum_{t \in [T]} \bar{r}_{\mathcal{D}}(a^*) - \bar{r}_{\mathcal{D}}(a_t)\right] &\leq \mathbb{E}\left[\sum_{t \in [T]} \bar{r}_{\widetilde{\mathcal{D}}}(a^*) - \bar{r}_{\widetilde{\mathcal{D}}}(a_t)\right] + \sum_{t \in [T]} \sqrt{\frac{m}{T}} \\
&= \mathbb{E}\left[\sum_{t \in [T]} \bar{r}_{\widetilde{\mathcal{D}}}(a^*) - \bar{r}_{\widetilde{\mathcal{D}}}(a_t)\right] + \sqrt{mT}.
\end{aligned} \tag{9}
$$

For the regret bound of the first term in the above, we treat $\widetilde{\mathcal{D}}$ as the true distribution. By following the proof steps in Theorem 5, we provide a bound for the regret as follows:

$$
\mathbb{E}\left[\sum_{t \in [T]} \bar{r}_{\widetilde{\mathcal{D}}}(a^*) - \bar{r}_{\widetilde{\mathcal{D}}}(a_t)\right]
$$

$$
= \mathbb{E}\left[\sum_{t \in [T]} (\bar{r}_{\widetilde{\mathcal{D}}}(a^*) - \bar{r}_{\widetilde{\mathcal{D}}}(a_t))\mathbb{1}(\mathcal{E}_t)\right] + \mathbb{E}\left[\sum_{t \in [T]} (\bar{r}_{\widetilde{\mathcal{D}}}(a^*) - \bar{r}_{\widetilde{\mathcal{D}}}(a_t))\mathbb{1}(\mathcal{E}_t^c)\right]
$$

$$
\leq \mathbb{E}\left[\sum_{t \in [T]} (\bar{r}_{\widetilde{\mathcal{D}}}(a^*) - \bar{r}_{\widetilde{\mathcal{D}}}(a_t))\mathbb{1}(\mathcal{E}_t)\right] + L \sum_{t \in [T]} \sum_{l=1}^{t-1} \sum_{i \in [d]} \mathbb{P}\left(\sup_{x \in [0,1]} |\hat{F}_{i,l}(x) - F_i(x)| \geq \sqrt{\frac{3 \ln t}{2l}}\right)
$$

$$
\lesssim \mathbb{E}\left[\sum_{t \in [T]} (\widetilde{r}_t^{UCB}(a^*) - \bar{r}_{\widetilde{\mathcal{D}}}(a_t))\mathbb{1}(\mathcal{E}_t)\right] + Ld
$$

$$
\lesssim \mathbb{E}\left[\sum_{t \in [T]} (\widetilde{r}_t^{UCB}(a_t) - \bar{r}_{\widetilde{\mathcal{D}}}(a_t))\mathbb{1}(\mathcal{E}_t)\right]
$$

$$
\lesssim \mathbb{E}\left[L \sum_{t \in [T]} \sum_{i \in a_t} \sqrt{\frac{\log T}{n'_{t,i}}}\right],
$$

where the second inequality is obtained from (a) in Lemma 9 and the last inequality comes from $\mathcal{E}_t$ and (b) in Lemma 9. For bounding the last term, we have

$$
\mathbb{E}\left[ L \sum_{t \in [T]} \sum_{i \in a_t} \sqrt{\frac{\log T}{n'_{t,i}}} \right]
$$

$$
= \mathbb{E}\left[ L \sum_{i \in [d]} \sum_{\tau \in [\tau_i(T)]} \sum_{t \in \mathcal{T}_i(\tau)} \sqrt{\frac{\log T}{n'_{t,i}}} \right]
$$

$$
\leq \mathbb{E}\left[ L \sum_{i \in [d]} \sum_{\tau \in [\tau_i(T)]} \sum_{t \in \mathcal{T}_i(\tau)} \sqrt{\frac{\log T}{|\mathcal{T}_i(\tau - 1)|}} \right]
$$

$$
= \mathbb{E}\left[ L \sum_{i \in [d]} \sum_{\tau \in [\tau_i(T)]} |\mathcal{T}_i(\tau)| \sqrt{\frac{\log T}{|\mathcal{T}_i(\tau - 1)|}} \right]
$$

$$
= \mathbb{E}\left[ L \sum_{i \in [d]} \sum_{\tau \in [\tau_i(T)]} \sqrt{\frac{Tm \cdot |\mathcal{T}_i(\tau - 1)|}{d}} \sqrt{\frac{\log T}{|\mathcal{T}_i(\tau - 1)|}} \right]
$$

$$
= \mathbb{E}\left[ L \sum_{i \in [d]} \tau_i(T) \right] \sqrt{\frac{Tm \log T}{d}}
$$

$$
\lesssim L \log \log(Tm/d) \sqrt{dmT \log T},
$$

where the first inequality is obtained from $n'_{t,i} \geq |\mathcal{T}_i(\tau - 1)|$ for $t \in \mathcal{T}_i(\tau)$, the second equality is obtained from the condition of updates in the algorithm, and the last inequality is obtained from (8). This concludes the proof with (9).

∎

## A.11 Additional Experiments

Here, we present additional experimental results for the covariance-adaptive algorithms and for the setting with a general reward function. The following results confirm that, consistent with the observations in the worst-case linear reward setting (Section 6), our algorithms achieve significantly reduced oracle usage and improved computational efficiency, while maintaining tight regret performance.

### A.11.1 Covariance-adaptive

Here, we present experiments (Figure 3) on covariance-adaptive frameworks under linear reward settings. The mean reward of each base arm is independently sampled from a uniform distribution over $[0, 1]$, with $d = 10$ base arms and cardinality constraint $m = 3$. The reward noise is correlated according to a $d \times d$ positive semi-definite covariance matrix $\Sigma$, constructed as $AA^\top + I_d$ with normalization, where $A \in \mathbb{R}^{d \times d}$ is a randomly generated matrix. The stochastic rewards are then sampled from a multivariate Gaussian distribution with the specified mean vector and covariance matrix $\Sigma$.

### A.11.2 General Reward

Next, we present experiments (Figure 4) on general (non-linear) reward settings with $d = 5$ and $m = 2$. For each arm $i \in [d]$, the reward is sampled from a discrete distribution supported on the finite set $\{0.2, 0.4, 0.6, 0.8, 1\}$. The probability distribution for each arm is generated as follows: one value is randomly assigned to each arm. Then the value is assigned a large probability mass of $0.99$, while the remaining values share the remaining $0.01$ probability mass equally. The reward for an action is defined as the square root of the sum of the sampled rewards from the selected arms.

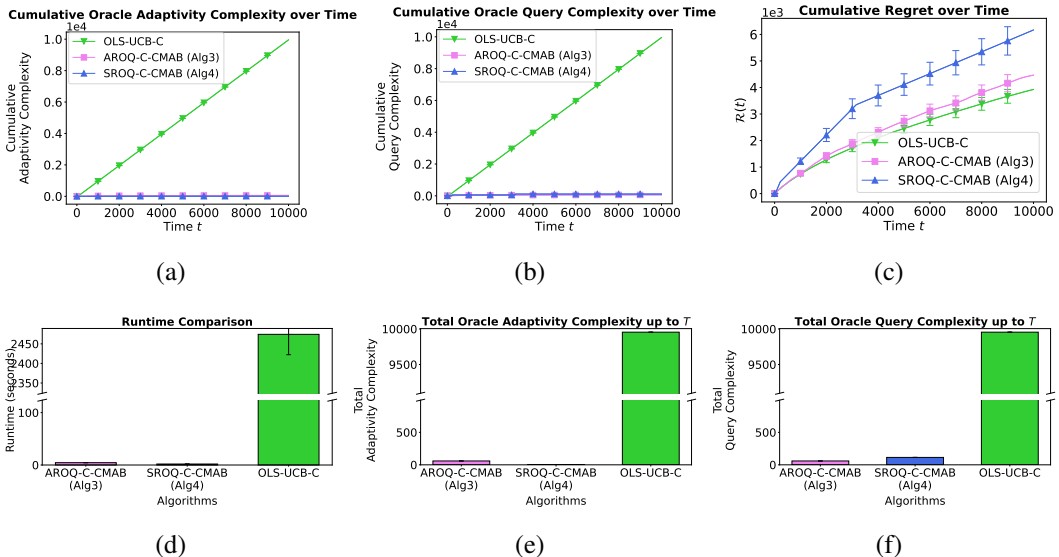

Figure 3: Experimental results for covariance-adaptivity under linear reward with $d = 10$ and $m = 3$: (a) cumulative oracle adaptivity complexity, (b) cumulative oracle query complexity, (c) regret, (d) runtime, (e) overall oracle adaptivity complexity, and (f) overall oracle query complexity of algorithms.

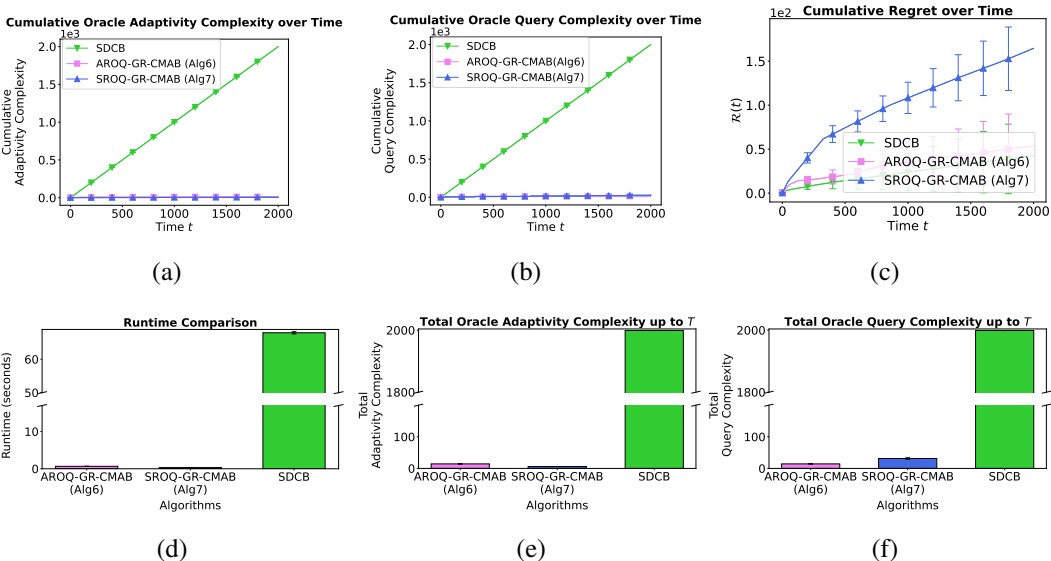

Figure 4: Experimental results for non-linear reward with $d = 5$ and $m = 2$: (a) cumulative oracle adaptivity complexity, (b) cumulative oracle query complexity, (c) regret, (d) runtime, (e) overall oracle adaptivity complexity, and (f) overall oracle query complexity of algorithms.

