# OpenReview forum: "Oracle-Efficient Combinatorial Semi-Bandits"
_NeurIPS.cc/2025/Conference — NeurIPS 2025 poster_

### Official Review · Reviewer_nCut · 2025-06-29

**Clarity:** 2
**Significance:** 3
**Originality:** 3
**Rating:** 5
**Confidence:** 3

**Summary:**

In this work, the authors consider the combinatorial semi-bandits under the spectrum of computation complexity.
Precisely, they consider a problem where at each round, a learner has to choose at most $m$ out of the $d$ available arms to play, and then observes a stochastic reward $y_i$ for each of the individual arms $i$ that were selected. These rewards are then combined into the global reward received by the learner. They mainly focus on the case where the reward function is a linear function of the arms ie $r(a_t, y_t) = <a, y_t>$ where $a_t$ is the binary vector indicating which of the $d$ arms are picked. (they also later generalize that notion to reward functions that are monotonous).
When the learner can make as many queries as they want, the problem is fairly well understood (see [16] in the authors references) where at each round of the game the learner can choose the subset of arms to play freely and [16] already provides a tight-up-to-log-factors regret bound of order $ O (\sqrt{mdT \log T})$ using the current paper's notations. The novelty of this work is that the authors propose novel algorithms that reduce the query complexity.

They consider two measures: adaptability complexity, where the learner essentially limits the number of rounds where the algorithm queries an optimisation oracle (several queries can happen in parallel and still count as 1) , and the query complexity where every single query is counted instead.

The authors propose two algorithms tailored for each of these measures, and then they provide a regret analysis as well as both query complexity measures for both algorithms.

The first algorithm, AROQ - CMAB achieves a $O( \sqrt{mdT \log T} \log\log(Tm/d))$ regret bound (which is only a $\log\log$ term over the bound of [16] while reducing the query complexity to $O(d\log\log(Tm/d))$ for both measures.
The second algorithm reduces the complexity further by removing the $d$ dependency of the total number of arms, but comes at the cost of a multiplicative $\sqrt m$ factor in the regret bound.

The authors provide detailed proofs in the appendix for all their results and experiments that compare the performance (both in regret and query cost) of their algorithm vs the non-query efficient baselines.

**Questions:**

See weaknesses:
Is there a reason why you don't also propose the regret bounds also expressed in a problem-dependent $\log T/\Delta$ type of regret?

**Ethical Concerns:**

["NO or VERY MINOR ethics concerns only"]

**Final Justification:**

I maintain my score and believe the paper should be accepted. The paper is interesting, and the comments mainly regarded the structure and clarity of the paper, which I am confident will be appropriately addressed in the final version.

**Limitations:**

yes.

**Paper Formatting Concerns:**

no issue.

**Quality:**

3

**Strengths And Weaknesses:**

This paper takes the problem of combinatorial bandits, which appears well understood and tackles the problem of query complexity in a new way. Their results seem to indicate that they would be useful in practice, as a big limitation of combinatorial semi-bandit problems is that the arm selection is an NP-hard problem, which can be very computationally expensive. While previous approaches might have considered using approximation algorithms for this optimisation, the authors' approach goes in a different direction of reducing the number of queries to this oracle rather than decreasing the quality (and complexity cost) of the oracle.

The paper is well written, the results are intuitive and the algorithms simple, which should make them easy to apply in practice.

As a potential weakness, it would be interesting to also get log-type regret bounds for this algorithm (that depend on sub-optimality gaps).
Some parts of the experiments (at least their description) have to be reworked, but I believe that this is relatively easily fixable.

In section 6, why do you use the algorithm of [6] in your baseline when the algorithm of [16] achieves better performance?
(Note that the algorithm of [6] is called CUCB, and CMAB is just the name of the problem.) This doesn't make for a fair comparison.
It would be interesting to take a slightly larger time horizon to compare the methods better, as your AROQ-CMAB has a better upper bound than CUCB when $m$ is large enough.
For the experiments in the appendix, it is even less clear which algorithms you are talking about. If the readers are not very familiar with the literature on these extensions of CMAB, they won't know where these algorithms are from.

If the time and space permit in the final version of the paper, it would be interesting to have different experiment ratios of $d$ and $m$ to get a better picture of the trade-offs (in particular, the query-cost vs regret between your two methods).

---

> ### Author Rebuttal · Authors · 2025-07-30
>
> We sincerely thank the reviewer for the thoughtful comments. Below, we provide point-by-point responses to each one.
>
>
> $\bullet$ **It would be interesting to also get log-type regret bounds for this algorithm.**
>
> **Response:** Thank you for your insightful comment. In this work, we focused on the gap-free setting as a foundational step toward oracle-efficient algorithms for combinatorial semi-bandits. Extending our frameworks to derive gap-dependent regret bounds is indeed a valuable direction for future work.
>
> One natural approach would be to incorporate elimination-based strategies, as commonly used in the batched multi-armed bandit literature [14]. However, in the combinatorial setting, elimination becomes more involved due to the need to construct representative actions for each base arm, as described in Line 3 of Algorithm 2. These representative actions allow us to eliminate large sets of suboptimal actions by targeting individual suboptimal base arms. However, pulling representative actions does not guarantee a uniform number of base-arm observations across epochs. This mismatch can introduce inefficiencies and prevent standard elimination techniques from achieving tight gap-dependent bounds, thereby limiting their direct applicability in our combinatorial setting.
>
> Nevertheless, we agree that developing gap-dependent variants within our oracle-efficient framework is a promising and important direction for future research.
>
>
>
> $\bullet$ **In section 6, why do you use the algorithm of [6] in your baseline when the algorithm of [16] achieves better performance? (Note that the algorithm of [6] is called CUCB, and CMAB is just the name of the problem.) This doesn't make for a fair comparison. It would be interesting to take a slightly larger time horizon to compare the methods better, as your AROQ-CMAB has a better upper bound than CUCB when
>  is large enough.**
>
> **Response:** Thank you for pointing this out. You are absolutely right; we should refer to the algorithm from [6] as CUCB, not CMAB, since CMAB refers to the problem setting rather than the algorithm itself. We will revise the terminology accordingly in the final version.
>
> Regarding the choice of baseline, all benchmark algorithms used in our experiments represent the state-of-the-art for each respective setting. For the linear reward setting,
>  both [6] and [16] analyze the same CUCB algorithm. In fact, [16] refers to it as CombUCB1 (see the second paragraph of their introduction), but the algorithm remains unchanged. This is why we use the same name for the algorithm in both [6] and [16] in Table 1. The main difference lies in the analysis: [16] provides a tighter bound than that in [6], establishing a near-optimal regret of $\widetilde{O}(\sqrt{mdT})$. In our work, AROC-CMAB (Algorithm 1) achieves the same near-optimal regret order (up to logarithmic factors), while significantly reducing oracle complexity -- which is our primary focus.
>
>
> We agree that evaluating longer horizons could further highlight the benefits of our algorithm.
>
> We will revise the terminology, clarify our choice of benchmarks, and elaborate on additional experiments in the final version.
>
>
>
> $\bullet$ **For the experiments in the appendix, it is even less clear which algorithms you are talking about. If the readers are not very familiar with the literature on these extensions of CMAB, they won't know where these algorithms are from.**
>
> **Response:** Thank you for the helpful comment. We agree that the appendix could be made clearer for readers who may not be deeply familiar with the literature on CMAB extensions. Regarding baselines, each representing the state of the art in its respective setting, we provide a brief summary here:
>
> - *Worst-case linear reward setting*: We use CUCB [6, 16] as the benchmark, which applies a standard UCB strategy by computing confidence bounds for each base arm and summing them over the selected actions. This algorithm was introduced in [6] and further analyzed in [16].
>
> - *Covariance-dependent setting*: We use OLS-UCB-C [26], a covariance-adaptive algorithm that estimates the noise structure online and selects actions optimistically based on this estimation.
>
> - *General reward setting*: We use SDCB [5], which estimates the reward distributions of individual arms and constructs stochastically dominant confidence bounds for action selection.
>
> We will revise the appendix in the final version to clearly reference the sources of these algorithms and provide brief descriptions to help guide the reader.
>
> $\bullet$ **If the time and space permit in the final version of the paper, it would be interesting to have different experiment ratios of $d$
>  and $m$
>  to get a better picture of the trade-offs (in particular, the query-cost vs regret between your two methods).**
>
> **Response:** Thank you for the constructive suggestion. We appreciate your point about exploring different ratios of $d$ (number of base arms) and $m$ (action size) to better understand the trade-offs between query cost and regret.
>
> We agree that varying these parameters could reveal interesting regime-dependent behaviors between our two algorithms. In particular, increasing $m$ may widen the regret gap between the methods, while increasing $d$ may amplify differences in oracle adaptivity complexity.
>
> For example, in additional experiments when increasing $d$ from 20 to 30, the synchronized algorithm of SROQ-CMAB (Algorithm 2) demonstrated a substantial advantage in oracle adaptivity complexity. The total adaptivity complexities were: SROQ-CMAB (Algorithm 2): 6, ARQO-CMAB (Algorithm 1): 73, and CUCB: 19,999 -- reflecting the theoretical improvements achieved by the synchronized update mechanism.
>
>
> When we increased $m$ from 3 to 5, the regret of SROQ-CMAB  became much larger than that of AROQ-CMAB  and CUCB. Specifically, the cumulative regret at time $T$ was: SROQ-CMAB: 1140, ARQO-CMAB: 678, and CUCB: 659 -- consistent with our theoretical result that the synchronized version incurs an additional $\sqrt{m}$ factor in regret.
>
>
>
> We will add such an ablation study in the final version.

---

> > ### Comment · Reviewer_nCut · 2025-08-06
> >
> > Thank you for the answers. My concerns have been addressed appropriately.

---

### Official Review · Reviewer_LFor · 2025-06-30

**Clarity:** 3
**Significance:** 3
**Originality:** 3
**Rating:** 4
**Confidence:** 3

**Summary:**

This paper studies the **combinatorial semi-bandit** problem, a generalization of the classical multi-armed bandit (MAB) framework in which an agent selects a subset of base arms and receives individual feedback for each selected arm. A major bottleneck in this setting is the **high computational cost of solving the underlying combinatorial optimization problem**, which is often NP-hard. Existing algorithms typically assume access to an **oracle** that must be queried at every round, resulting in high adaptivity and query complexity.

To address this, the authors propose two novel **oracle-efficient algorithmic frameworks** that significantly reduce oracle usage while retaining strong theoretical guarantees:

1. **Adaptive Rare Oracle Queries (AROQ)** – adaptively schedules oracle queries to reduce invocation frequency.
2. **Synchronized Rare Oracle Queries (SROQ)** – synchronizes and batches oracle queries at fixed epochs, allowing for parallelism and even lower adaptivity.

The frameworks are instantiated and analyzed under three reward models:

- Worst-case linear rewards,
- Covariance-adaptive linear rewards,
- General non-linear reward models.

Theoretical results show that the proposed algorithms **reduce oracle usage from linear to (doubly) logarithmic** in time $T$, while achieving near-optimal regret bounds. These results are supported by empirical evaluations on synthetic datasets.

**Questions:**

Please comment on or discuss the above weakness points.

**Ethical Concerns:**

["NO or VERY MINOR ethics concerns only"]

**Final Justification:**

I believe this paper is solid and is valuable for advancing the field of efficient combinatorial MAB, so I vote for acceptance of the current paper.

**Limitations:**

Yes.

**Paper Formatting Concerns:**

No.

**Quality:**

3

**Strengths And Weaknesses:**

### **Strengths**

1. **Addresses a Critical Practical Challenge**: The work tackles the key limitation of **excessive oracle invocation** in combinatorial semi-bandits, which severely limits scalability. The paper clearly identifies this as a computational bottleneck and proposes a principled solution.
2. **Novel Oracle-Efficient Frameworks**: The proposed **AROQ** and **SROQ** frameworks represent an important methodological innovation. While adaptive querying has been explored in other bandit settings, this is the first work (to the authors’ knowledge) to explicitly target **oracle efficiency** in the **combinatorial semi-bandit** setting. The two approaches also offer flexibility: AROQ is simple and adaptive, while SROQ supports **parallelization** via synchronized updates.
3. **Strong and General Theoretical Results**: The paper provides regret guarantees that match known lower bounds (up to logarithmic factors), while simultaneously reducing oracle usage:
    - For **worst-case linear rewards**, the AROQ-CMAB algorithm achieves regret $\tilde{O} (\sqrt{mdT})$ with oracle query complexity $O(d \log \log(Tm/d))$, a significant improvement over the standard $\Theta(T)$ baseline.
    - The SROQ-CMAB algorithm further reduces **oracle adaptivity** to $\Theta(\log \log T)$, while maintaining the same query complexity.
    - For **covariance-adaptive linear rewards**, AROQ-C-CMAB achieves query complexity $O(d^2 \log(Tm))$, and SROQ-C-CMAB maintains $\Theta(\log \log T)$ adaptivity.
    - For **general reward models**, AROQ-GR-CMAB achieves $O(d \log \log(Tm/d))$ query complexity and SROQ-GR-CMAB again attains $\Theta(\log \log T)$ adaptivity.

    These results are substantial as they show the feasibility of low-adaptivity learning even in high-dimensional, structured settings.

4. **Broad Applicability and Model Generality**: The proposed methods are not limited to a specific reward structure. Extensions to **covariance-aware** and **general non-linear** reward models greatly enhance the framework’s **practical scope**, making it applicable to domains like recommender systems, online resource allocation, and network routing.


---

### **Weaknesses / Limitations**

1. **Gap-Dependent Regret Bounds Not Discussed**: The paper only presents **gap-free** regret bounds. It would strengthen the contribution to investigate whether the same frameworks can be extended to derive **gap-dependent** bounds, which often provide **tighter** and **problem-dependent** regret characterizations, especially in practical settings.
2. **Experimental Evaluation is Limited**: The empirical results are based solely on **small-scale synthetic datasets**. There is no validation on **real-world datasets**, which weakens the practical evaluation. It would be valuable to include experiments on datasets from real applications (e.g., recommendation, online ads, network routing) to demonstrate scalability and generalization.
3. **Minor Presentation Issues**:
    - In **line 140**, the proof of **Theorem 1** is said to be in Appendix B.2, but the citation appears to be off.
    - In **line 175**, the proof of **Theorem 2** is listed incorrectly; it should refer to Appendix B.4.
    - In **line 174**, the adaptivity complexity uses $\Theta(\log \log T)$; However, there is no lower bound showing lower bound results.

---

> ### Author Rebuttal · Authors · 2025-07-30
>
> We sincerely thank the reviewer for the thoughtful comments. Below, we provide point-by-point responses to each one.
>
> $\bullet$ **Discussion on Gap-Dependent Regret Bounds**
>
>
> **Response:** Thank you for your insightful comment. In this work, we focused on the gap-free setting as a foundational step toward oracle-efficient algorithms for combinatorial semi-bandits. Extending our frameworks to derive gap-dependent regret bounds is indeed a valuable direction for future work.
>
> One natural approach would be to incorporate elimination-based strategies, as commonly used in the batched multi-armed bandit literature [14]. However, in the combinatorial setting, elimination becomes more involved due to the need to construct representative actions for each base arm, as described in Line 3 of Algorithm 2. These representative actions allow us to eliminate large sets of suboptimal actions by targeting individual suboptimal base arms. However, pulling representative actions does not guarantee a uniform number of base-arm observations across epochs. This mismatch can introduce inefficiencies and prevent standard elimination techniques from achieving tight gap-dependent bounds, thereby limiting their direct applicability in our combinatorial setting.
>
> Nonetheless, we consider the development of gap-dependent oracle-efficient algorithms an exciting and important direction for future work.
>
>
> $\bullet$ **There is no validation on real-world datasets.**
>
> **Response:** Thank you for the thoughtful comment. We appreciate the importance of real-world validation, particularly in application-driven domains. However, in the bandit literature, particularly in combinatorial and semi-bandit settings,  the primary focus is on theoretical contributions [6, 5, 16, 10, 20, 26]. Even when experiments are included, they are typically conducted on controlled synthetic environments to precisely validate regret behavior and algorithmic properties.
>
> Our main objective in this work is to investigate oracle efficiency in combinatorial semi-bandits from a theoretical standpoint. Accordingly, our synthetic experiments are conducted to support and illustrate the theoretical guarantees, in line with standard practice in the literature.
>
> While real-world datasets are outside the scope of this work, we believe that incorporating domain-inspired structures (e.g., recommendation or routing) could be an interesting extension for future empirical validation.
>
> $\bullet$ **Minor Presentation Issues: In line 140, the proof of Theorem 1 is said to be in Appendix B.2, but the citation appears to be off. In line 175, the proof of Theorem 2 is listed incorrectly; it should refer to Appendix B.4.**
>
> **Response:**
> Thank you for your helpful comments. You are correct - the appendix references for the proofs of Theorems 1 and 2 were mistakenly interchanged. We will correct these citations to ensure that each theorem properly points to its corresponding appendix section.
>
>
>
> $\bullet$ **Minor Presentation Issues: In line 174, the adaptivity complexity uses  $\Theta(\log\log T)$; However, there is no lower bound showing lower bound results.**
>
> **Response:** Thank you for pointing this out. To clarify, the oracle adaptivity complexity of $\Theta(\log\log T)$ in Theorem 2 is not derived from a lower-bound analysis, but rather corresponds to the *explicit number of synchronized update steps* used by Algorithm 2. Specifically, the algorithm sets the number of updates to $M = \Theta(\log\log T)$, as stated in Section 3.2 (Lines 170-173). As shown in Lines 763-766 of the appendix, since each update requires a round of oracle queries, the total adaptivity complexity directly follows from this setting of $M$. For a more detailed explanation regarding synchronized updates, please refer to Appendix B.1.
>
> We will revise the main text to better explain this connection and avoid confusion regarding the interpretation of this complexity term.

---

> > ### Comment · Reviewer_LFor · 2025-08-06
> >
> > Thanks for the authors’ feedback. I maintain my positive position after reviewing their response.

---

### Official Review · Reviewer_RoU1 · 2025-07-07

**Clarity:** 2
**Significance:** 3
**Originality:** 3
**Rating:** 4
**Confidence:** 4

**Summary:**

The paper considers the problem of minimizing the queries to an oracle to find the optimal arm to play at every round in a combinatorial semi-bandit setup. The oracle calls can be costly as the oracle needs to find the optimal arm from a set which is exponentially large in the number of base arms. For this setup, the authors use an interesting insight where the confidence intervals of the arms rewards do not change by $O(1/\sqrt{\mathcal{T}_i}(\tau_i))$ where $\mathcal{T}_i(\tau_i)$ is the number of times arm $i$ is played in epoch $\tau_i$ and the regret is upper bounded by the length of the next epoch times the gap in the current round. With this insight, the authors are able to bound the calls to the computation oracle to switch the policy or the optimal arm to play in the epoch, while still maintaining low regret. The authors then extend the findings to parallelize oracle calls for each base arms and further reduce the adaptive calls.

**Questions:**

None

**Ethical Concerns:**

["NO or VERY MINOR ethics concerns only"]

**Final Justification:**

Authors have added details regarding experiment evaluations which was my major criticism.

**Quality:**

2

**Strengths And Weaknesses:**

Strengths:

1. The use of bounding the algorithm regret by the confidence interval gaps of previous round for combinatorial bandits is interesting.

2. The parallelization of oracle calls to further reduce policy switches is interesting and novel.


Weakness:

1. Algorithm 2 takes in the pre-determined sequence of policy updates. What happens if the pre-determined sequence is mis-configured? Ideally, the algorithm should determine the sequence.

2. Weak evaluations: Authors must discuss the findings of the evaluation results in details and must elaborate why the algorithms behave as they do. Also, in the appendix, new algorithms are introduced without description. It is a good practice to remind the reader what the baseline algorithms are and how do they work.

3. I think the authors have incorrect references. The proof of Theorem 1 maps to the proof of Theorem 2.

---

> ### Author Rebuttal · Authors · 2025-07-30
>
> We sincerely thank the reviewer for the thoughtful comments. Below, we provide point-by-point responses to each one.
>
>
> $\bullet$ **Algorithm 2 takes in the pre-determined sequence of policy updates.**
>
>
>
> **Response:**  You are correct that Algorithm 2 relies on a pre-determined sequence of policy updates. This design choice is intentional and central to our goal of reducing oracle adaptivity complexity: it enables synchronized oracle queries that further reduce adaptivity complexity, even beyond what is achieved by Algorithm 1. Reducing oracle adaptivity is essential to reduce computation cost, and using pre-scheduled, infrequent updates aligns with prior research on batched bandit algorithms [14, 19], which inspired our framework as discussed in the main text.
>
> In practice, a fixed update schedule is rarely a significant limitation. In many real-world applications, computational budgets and synchronization constraints are known in advance, making it natural to plan update schedules accordingly. Moreover, as shown in Figures 2, 3, and 4, our approach with synchronized updates achieves the best runtime performance compared to other algorithms.
>
> Developing an adaptive version of the synchronized update scheme could be an interesting direction for future research. We will include this discussion in the final version.
>
>
>
>
> $\bullet$ **Authors must discuss the findings of the evaluation results in details and must elaborate why the algorithms behave as they do.**
>
> **Response:** Thank you for your helpful feedback. We agree that interpreting evaluation results is important, and we are happy to elaborate further.
>
> As predicted by our theoretical analysis, the proposed algorithms significantly reduce oracle complexity through rare update policies. This is clearly demonstrated in Figure 2(a) and 2(b), where the oracle query and adaptivity complexities of baseline methods grow linearly with time, whereas our algorithms exhibit sublinear growth. This results in substantial computational savings, as shown by the faster runtime in Figure 2(d), and reduced total oracle usage in Figures 2(e) and 2(f).
>
> As expected, the infrequent updates result in slightly higher regret in the early rounds (Figure 2(c)). However, as more information is accumulated, the performance of our algorithms improves and eventually approaches that of the baseline, while maintaining significantly better oracle efficiency. This improvement occurs as the cumulative information gained over time mitigates the impact of rare updates. Similar trends are also observed in the covariance-dependent and general reward settings (Appendix B.11).
>
> We will clarify these points and expand the discussion in the final version to better guide the reader through the observed behavior.
>
>
>
> $\bullet$ **In the appendix, new algorithms are introduced without description.**
>
> **Response:**  Thank you for your comments. We clarify that each algorithm introduced in the appendix is referenced and motivated in the main text.
>
> Specifically, Algorithm 5 in Appendix B is introduced in connection with the discussion of $\alpha$-approximation oracles in the last paragraph of Section 3.1 of the main text (Lines 147-154). Algorithms 6 and 7 in Appendix B correspond to our oracle-efficient methods for the general reward setting, as indicated in Section 5 of the main text (Lines 255-258). Furthermore, Algorithm 8 in Appendix B formalizes our extension from discrete to continuous reward distributions, which is referenced in footnote 2 of Section 5.
>
> We will make these cross-references more explicit in the final version to improve clarity and readability.
>
>
>
> $\bullet$ **It is a good practice to remind the reader what the baseline algorithms are and how do they work.**
>
> **Response:** Thank you for your suggestion. Regarding baselines, each representing the **state-of-the-art** in its respective setting, we provide a brief summary here:
>
> - *For the worst-case linear reward setting*, CUCB [6, 16]   serves as our benchmark. It uses a standard UCB strategy by constructing upper confidence bounds for each base arm and summing them over actions.
>
> - *For the covariance-dependent setting*, we use OLS-UCB-C [26] as a benchmark, which is a covariance-adaptive algorithm that estimates the noise structure online and selects actions optimistically based on this estimation.
>
> - *For the general reward setting*, SDCB [5] is the benchmark. It estimates the distributions of the reward variables and constructs stochastically dominant confidence bounds for action selection.
>
> A key difference is that all baseline methods require an oracle call at every round, whereas our algorithms significantly reduce oracle usage while maintaining competitive regret. We will clarify these baselines and their connections to our methods in our final version.
>
>
> $\bullet$ **I think the authors have incorrect references. The proof of Theorem 1 maps to the proof of Theorem 2.**
>
>
> **Response:** Thank you for pointing this out. You are correct. The references to the appendix for the proofs of Theorems 1 and 2 were mistakenly interchanged. We will revise the text to ensure that each theorem correctly points to its corresponding appendix section.
>
> ---
> **Overall**: We sincerely appreciate the reviewer’s constructive feedback, which has helped us clarify and improve our work. We hope that the responses and clarifications above address the concerns raised, and we would be grateful if the reviewer would consider them in their final evaluation. We would be happy to provide further clarifications if needed.

---

> > ### Author Response · Authors · 2025-08-06
> >
> > Thank you again for the thoughtful and constructive feedback. We hope that our rebuttal, especially the clarifications regarding the algorithmic design choices, interpretation of experimental results, explanations of baseline methods and appendix algorithms, as well as the correction of reference errors, has addressed your concerns.
> >
> > If any part remains unclear or could benefit from further elaboration, we would be happy to provide additional clarification.

---

> > > ### Comment · Reviewer_RoU1 · 2025-08-06
> > >
> > > Thank you for the answers. My concerns have been addressed appropriately and based on that I have increased my ratings.

---

### Note · Authors · 2025-08-11

We sincerely thank the Reviewers and the Area Chair for their thoughtful and constructive feedback. We are pleased that our responses have helped address the concerns raised during the review process, and we will incorporate the improved explanations, additional experiments, and necessary minor edits into the final version of the paper.

Our work presents oracle-efficient algorithms for combinatorial semi-bandits with strong theoretical guarantees and significantly reduced oracle complexity.

---

### Decision · Program_Chairs · 2025-09-17

**Decision:**

Accept (poster)

**Comment:**

The authors study the combinatorial bandit problem with semi-bandit feedback. They demonstrate the existence of an algorithm that achieves a near-optimal minimax regret bound while only requiring a number of argmax oracle calls that scales like loglog(T). This is significant because some combinatorial optimizations made not be efficient, thus making it desirable to minimize the number of calls. The reviewers found the algorithm design to be novel and the results to be a substantial contribution to the area.